# A Taxonomic Review of the Genus *Telsimia* Casey (Coleoptera, Coccinellidae) from China, with Descriptions of Eight New Species

**DOI:** 10.3390/insects13100869

**Published:** 2022-09-24

**Authors:** Keke Bi, Lizhi Huo, Xingmin Wang

**Affiliations:** 1Guangzhou Institute of Forestry and Landscape Architecture, Guangzhou 510642, China; 2Engineering Technology Research Center of Agricultural Pest Biocontrol, College of Plant Protection, South China Agricultural University, Guangzhou 510642, China

**Keywords:** Coccinellinae, Chilocorinae, Telsimiini, natural enemy, biological control

## Abstract

**Simple Summary:**

The *Telsimia* species are known as important predators of diaspidine scale insects that infest citrus, coconut, camphor, and bamboo plants, among others. A clear understanding of *Telsimia* species diversity and their geographical distribution forms the basis for the introduction and utilization of these natural enemies. At present, 48 species (subspecies) are known around the world, and studies on Chinese *Telsimia* species are relatively scarce. In this paper, 20 species of the genus *Telsimia* from China are reviewed, including eight new species and two new records. This taxonomic study can provide a theoretical basis for the development and utilization of *Telsimia* species for biological control of diaspidine scale insects.

**Abstract:**

Twenty species of the genus *Telsimia* from China are reviewed herein. Among them, eight species are described as new to science: *Telsimia chayuensis*, *T.* *forcipata*, *T.* *latus*, *T.* *lobatus*, *T.* *lunata*, *T.* *menglaensis*, *T.* *parascymnoides*, and *T.* *parvus* spp. nov.; two species are reported from China for the first time: *T.* *darjeelingensis* Kapur, 1969 and *T.* *elongate* Hoàng, 1985. All species are provided with nomenclatural history, diagnoses, detailed descriptions (except for the 10 previously described species), colored illustrations, and distributions. The female genitalia of five described species are provided for the first time. A distribution map and a key to all the Chinese species are also provided.

## 1. Introduction

The genus *Telsimia* Casey, 1899 [1] belongs to the tribe Telsimiini of the subfamily Coccinellinae [2]. It is known in Africa and many parts of Asia and is widespread in the Pacific Islands, New Guinea, and Australia [3]. The *Telsimia* species are known as important predators of diaspidine scale insects that infest citrus, coconut, camphor, and bamboo plants, among others [4,5,6,7,8]. For instance, certain *Telsimia* species were introduced from their original places into Micronesia, Hawaii, and California, and efficient biological control of coconut scale *Aspidiotus destructor* Signoret and citrus scale *Aonidiella aurantii* (Maskell) was achieved [9,10,11,12]. A clear understanding of *Telsimia* species diversity and their geographical distribution forms the basis for the introduction and utilization of these natural enemies. This taxonomic study can provide a theoretical basis for the development and utilization of *Telsimia* species for the biological control of diaspidine scale insects. 

At present, 48 species (subspecies) are known around the world, while 10 species (subspecies) are known in China (Appendix A) [1,3,13,14,15,16,17,18,19,20,21,22,23,24,25,26,27,28,29,30,31,32,33,34,35,36,37,38,39]. No new species have been described from China since 1979, when Pang and Mao published four new species (subspecies) [33]. With the accumulation of ladybug specimens in recent years, it is necessary to carry out more taxonomic studies on Chinese *Telsimia* species. 

In this paper, by studying the specimens of *Telsimia* preserved in the Insect Collection of South China Agricultural University, eight new species and two new Chinese records are found; 10 previously described species are also reviewed. Nomenclatural history, diagnoses, detailed descriptions (except for the 10 previously described species), colored illustrations, and distributions are provided for each species. The female genitalia of five described species are provided for the first time. A distribution map and a key to all the Chinese species are also provided.

## 2. Materials and Methods

All studied materials were preserved in the Insect Collection of South China Agricultural University (SCAU) except for three holotypes from the Entomological Laboratory, Ehime University, Matsuyama, Japan (EUMJ). Morphological terms follow Ślipiński [2] and Ślipiński and Tomaszewska [40]. External morphology was observed using a stereomicroscope (SteREO Discovery V20, Zeiss). Measurements were carried out using an ocular micrometer attached to the stereomicroscope and are defined as follows: (TL) total length, from apical margin of clypeus to apex of elytra; (TW) total width, across both elytra at widest part; (TH) total height, through the highest point of elytra to metaventrite; (HW) head width, including eyes; (PL) pronotal length, from the middle of anterior margin to the base of pronotum; (PW) pronotal width at widest part; (EL) elytral length, along the suture, from the apex to the base, including the scutellum; (EW) elytral width, across both elytra at widest part; (ID) interocular distance, nearest distance between two eyes. Male and female genitalia were dissected, cleared by boiling in 10% NaOH solution for several minutes, and examined using a compound microscope (ZEISS Imager M2).

Images of adults were photographed using a digital camera (EOS 5DSR, Canon) and lens (MP-E 65 mm f/2.8 1–5×, Canon). The camera was mounted on a focus stacking rail (WeMacro Rail) and controlled with the software Helicon Remote v3.9.12. A series of images were stacked using the software Helicon Focus v8.1.1 (HeliconSoft). Images of genitalia were photographed using a digital camera (Axiocam 506 color) connected to a microscope (ZEISS Imager M2). The software ZEN 2.3 was used to capture images. Images were cleaned up and laid out in plates using Adobe Photoshop CC 2019.

The order of species names is in the order of the key. Most labels of the specimens examined in this study are in Chinese, which have beencorrespondingly translated into English. In terms of geographical distribution, country names are sorted alphabetically. The Chinese provinces are sorted according to their official order: North China: Beijing, Tianjin, Hebei, Shanxi, Inner Mongolia; Northeast China: Liaoning, Jilin, Heilongjiang; East China: Shanghai, Jiangsu, Zhejiang, Anhui, Fujian, Jiangxi, Shandong; Central China: Henan, Hubei, Hunan; South China: Guangdong, Guangxi, Hainan; Southwest China: Chongqing, Sichuan, Guizhou, Yunnan, Tibet; Northwest China: Shaanxi, Gansu, Qinghai, Ningxia, Xinjiang; special administrative regions or special status areas: Hong Kong, Macao, Taiwan.

## 3. Results

### 3.1. Taxonomy


***Telsimia scymnoides* Miyatake, 1978 小毛寡节瓢虫**


(Figure 1 and Figure 2)

*Telsimia scymnoides* Miyatake, 1978: 15 [23]; Yu, 1995: 141 [35]; Ren et al., 2009: 152 [39].

**Diagnosis.** This species can be easily distinguished from other *Telsimia* species by its unique coloration: head and pronotum entirely brownish yellow; elytra black, each elytron with a large, oblique, oval, brownish yellow spot at apical half, reaching lateral margin, suture, and apex (Figure 1a–c). It can also be diagnosed by its male genitalia: in ventral view, base of penis guide is nearly the same width as phallobase, slightly convergent to apical one-fifth, then suddenly narrowing to a bifid apex (Figure 1j). 

**Material examined (16). Holotype:** [Taiwan], Sungkan Meifeng, Nantow Hsien, 25-26.V.1972, M. Sakai leg. //Holotype, *Telsimia scymnoides* M. Miyatake//1104 (♂, EUMJ). **Other materials:** China: **Guangdong Prov.**: Shimentai National Nature Reserve: 9♂2♀, Huangdong Village, 30.X.2004, Wang XM leg; 3♀, Wulangzhang Village, 8.X.2004, Wang XM leg; **Taiwan Prov.**: 1♂, Jiayi County, Fenchihu Lake, 1400m, 9.IV.1977, J. Klapperich leg, SCAUI11025. 

**Distribution.** China: Guangdong, Taiwan. 

**Biology.** Unknown. 


***Telsimia parascymnoides* Wang sp. nov. 拟小毛寡节瓢虫**


(Figure 2 and Figure 3)

**Etymology**. The specific epithet refers to its general appearance and penis guide closely resembling *T. scymnoides* Miyatake.

**Diagnosis.** In male genital structure, this species is most closely related to *T. scymnoides* but can be easily distinguished from the latter by detailed characteristics of penis guide: in ventral view, penis guide slightly convergent to apical one-sixth, then relatively smoothly narrowing to the apex, which is protruding and arcuate, with a tiny notch at middle (Figure 2g), while in *T. scymnoides*, penis guide slightly convergent to apical one-fifth, then suddenly narrowing to a bifid apex (Figure 1j).

**Description.** TL: 1.70 mm, TW: 1.30 mm, TH: 0.80 mm, PL: 0.48 mm, PW: 0.90 mm, HW: 0.56 mm, ID: 0.24 mm, TL/TW: 1.31, PL/PW: 0.53, EL/EW: 1.00, HW/PW: 0.62, PW/EW: 0.69; ID/HW: 0.43. 

Body oblong oval, convex, broadest at or very slightly behind middle; elytra entirely black except apical part brownish yellow; pronotum, head and clypeus brownish yellow (Figure 3a–c); mesoventrite and metaventrite dark brown; elytral epipleuron brownish yellow; mouthpart, prothoracic hypomeron, prosternum, and legs brownish yellow. Head between eyes 0.43 times the width of head; pubescence dense, appressed, and silver-white. Pronotum approximately twice as broad as median length; anterior border arcuate medially, distance between anterior angles 0.65 times that between posterior angles, lateral borders arcuately convergent from obtuse posterior to briefly rounded anterior angles; pubescence sparse, suberect, and silver-white. Elytra slightly longer than greatest combined breadth; external margins not explanate to borders; pubescence sparse, suberect, and silver-white. Abdominal postcoxal lines of ventrite 1 slightly recurved, punctures between coxae large and dense; ventrite 5 in midline as long as ventrites 2–4 together, apical border evenly arcuate in male (Figure 3d).

Male genitalia: Penis robust (Figure 3f), in lateral view, as long as tegmen, basal one-third gradually narrowing from the middle to distal end, which is thread-like; apical one-fifth gradually narrowing to a rounded apex. In lateral view of penis guide (Figure 3e), tegminal strut short, nearly one-fourth the length of tegmen; penis guide slender, gradually narrowing to a pointed apex, bent outward; parameres slender, slightly shorter than penis guide. In ventral view of penis guide (Figure 3g), phallobase nearly trapezoid, the top 1.5 times the length of the bottom; base of penis guide nearly the same width as base of phallobase, penis guide slightly convergent to apical one-sixth, then relatively smoothly narrowing to the apex, which is protruding and arcuate, with a tiny notch at middle.

Female genitalia: Unknown.

**Types. Holotype:** ♂, China: Yunnan Prov.: Mengla County, Yaoqu Town, ca. 700 m, 7–8.V.2009, Ren SX et al. leg, SCAU(E)16766. 

**Distribution.** China: Yunnan.

**Biology.** Unknown.


***Telsimia menglaensis* Wang sp. nov.**
**勐腊寡节瓢虫**


(Figure 2 and Figure 4) 

**Etymology.** The specific name refers to the locality of its holotype.

**Diagnosis.** In external appearance, it is similar to *T*. *parascymnoides* sp. nov., with the head, pronotum, and elytra terminal brownish yellow (Figure 4a–c), but can be easily distinguished from the latter by the male genitalia (Figure 3e–g and Figure 4g–i). The male genitalia are also closely related to *T*. *shirozui* but can be distinguished from the latter by the following characteristics: in ventral view of tegmen, base of phallobase half the width of proximal end, apical emargination of penis guide larger, inverted U shape; in *T*. *shirozui*, base of phallobase slightly narrower than proximal end, apical emargination smaller, inverted V shape. In lateral view of tegmen, penis guide thick, nearly parallel for the basal two-thirds, apical part sharper; in *T*. *shirozui*, penis guide relatively thinner, gradually narrowing to apical one-fourth, apical part blunt. 

**Description.** TL: 1.60–1.80 mm, TW: 1.24–1.44 mm, TH: 0.82–0.90 mm, PL: 0.44–0.50 mm, PW: 0.90–1.02 mm, HW: 0.56–0.64 mm, ID: 0.22–0.28 mm, TL/TW: 1.25–1.29, PL/PW: 0.49–0.51, EL/EW: 0.97–1.00, HW/PW: 0.60–0.63, PW/EW: 0.71–0.75, ID/HW: 0.39–0.44. 

Body oval, convex, broadest at middle; elytra black to crater brown, except apical part brownish yellow; pronotum, head and clypeus brownish yellow; mesoventrite and metaventrite dark brown; mouthparts, prothoracic hypomeron, prosternum, elytral epipleuron, and legs brownish yellow. Head between eyes nearly 0.4 times the width of head; pubescence dense, appressed, and silver-white. Pronotum approximately twice as broad as median length; anterior border arcuate medially, distance between anterior angles 0.6 times that between posterior angles, lateral borders arcuately convergent from obtuse posterior to briefly rounded anterior angles; pubescence sparse, suberect, and silver-white. Elytra slightly shorter than greatest combined breadth; external margins not explanate to borders; pubescence sparse, suberect, and silver-white. Abdominal postcoxal lines of ventrite 1 not recurved, punctures between coxae large and dense; ventrite 5 in midline as long as ventrites 2–4 together, apical border slightly notched medially in male and arcuate in female (Figure 4d,f).

Male genitalia: Penis, in lateral view (Figure 4g), as long as the tegmen, basal half distinctly narrower, apical one-fifth gradually narrowing to rounded apex. In lateral view of tegmen (Figure 4h), tegminal strut short, nearly one-sixth the length of tegmen; penis guide thick, thickness almost equals the width of penis guide in ventral view, nearly parallel for the basal two-thirds, apical part sharper; parameres distinctly shorter than penis guide. In ventral view of tegmen (Figure 4i), the base of phallobase half as wide as the proximal end, penis guide nearly parallel from base to apical one-fifth, then gradually narrowing to a bifid apex, the apical emargination larger, inverted U shape.

Female genitalia: As shown in Figure 4e. 

**Types (4)**. **Holotype**: ♂, China: **Yunnan Prov.**: Mengla County, Yaoqu Town, ca. 700 m, 7–8.V.2009, Ren SX et al. leg, SCAU(E)16769. **Paratypes (3)**: 1♂, with same data as holotype, SCAU(E)16781; 1♂1♀, **Guangxi Prov.**: Jingxi County, Huadong Town, Tongling Canyon, 5.VIII.2005, Ren SX et al. leg, SCAU(E)13965 and SCAU(E)13890.

**Distribution.** China: Guangxi, Yunnan.

**Biology.** Unknown.


***Telsimia latus* Wang sp. nov. 宽叶寡节瓢虫**


(Figure 2 and Figure 5) 

**Etymology.** The specific epithet is the Latin adjective “*latus*”, meaning broad, referring to its broad penis guide in ventral view.

**Diagnosis.** Male genitalia of this species are similar to *T*. *daclacensis* Hoàng, 1983 with proximal end of penis guide in ventral view distinctly stretched out laterally. The difference is, in *Telsimia latus*, the outline of the protruding part on each side is similar to an arc (Figure 5g), while, in *T*. *daclacensis*, the protruding part on each side forms a downward projection (Hoàng 1983, p.7, Figure 2). 

**Description.** TL: 1.84 mm, TW: 1.48 mm, TH: 0.84 mm, PL: 0.50 mm, PW: 1.04 mm, HW: 0.62 mm, ID: 0.26 mm, TL/TW: 1.24, PL/PW: 0.48, EL/EW: 0.97, HW/PW: 0.60, PW/EW: 0.70, ID/HW: 0.42. 

Body oval, convex, broadest at middle; above black except anterior corners of pronotum and head brownish yellow (Figure 5a–c); underside dark brown to black, except elytral epipleuron brownish yellow, mouthparts, prothoracic hypomeron, and legs brownish yellow. Head between eyes nearly 0.4 times the width of head; pubescence dense, appressed, and silver-white. Pronotum slightly more than twice as broad as median length; anterior border arcuate medially, distance between anterior angles 0.6 times that between posterior angles, lateral borders arcuately convergent from obtuse posterior to briefly rounded anterior angles; pubescence dense, suberect, and silver-white. Elytra slightly shorter than greatest combined breadth; external margins not or very slightly explanate to borders; pubescence dense, suberect, and silver-white. Abdominal postcoxal lines of ventrite 1 slightly recurved, punctures between coxae small and sparse; ventrite 5 in midline as long as ventrites 2–4 together, apical border briefly notched medially in male (Figure 5d). 

Male genitalia: Penis thick (Figure 5e), curved at basal 1/3, as long as the tegmen. Tegminal strut extremely short (Figure 5f). Phallobase inflated. Parameres 1.5 times of phallobase length and 0.7 times of penis guide length, apical half with several long setae. Penis guide in lateral view gradually narrowing to a pointed apex. Penis guide tubular with a long fusiform opening on ventral side. Penis guide in ventral view 2.5 times as long as width of basal part, slightly narrowing to middle part, then expanded laterally, then convergent to a small blunt apex (Figure 5g). 

Female genitalia: Unknown. 

**Types. Holotype:** ♂, CHINA: **Hainan Prov.**: Yinggeling National Nature Reserve, 6.V.2005, Wang XM leg. No.20051211228, SCAU(E)16741.

**Distribution.** China: Hainan. 

**Biology.** Unknown.


***Telsimia elongate* Hoàng, 1985 狭长寡节瓢虫**


(Figure 2 and Figure 6) 

*Telsimia elongate* Hoàng, 1985: 41 [31].

**Diagnosis.** This species can be easily distinguished from other *Telsimia* species by its male genitalia: in lateral view (Figure 6h), tegmen bent nearly 90° at the middle part; in ventral view (Figure 6i), tegminal strut nearly triangular, broad at base, gradually narrowing to terminal, penis guide nearly parallel for the basal two-thirds, then gradually narrowing to a pointed apex. 

**Materials examined (30). China**: **Guangdong Prov.**: Shimentai National Nature Reserve: Huangdong Village, Wang XM leg: 6♂6♀, 30.X.2004; 2♂2♀, 31.X.2004; 1♂, Yangqiu Mountain, 6.X.2004, Wang XM leg; 1♂, Yangshan County, Chengjia Village, 9-14.VII.1997, Tian MY leg; 1♂, Dinghushan National Nature Reserve, 17.V.1989, Ren SX leg. **Guangxi Prov.**: 3♂2♀, Mao’er Mountains, Shili Canyon, 19.X.2004; Shiwandashan National Forest Park: 1♂, Hongqi Forest Farm, 11.XI.2004, Lv XB leg; 1♂, Fulong Mountains, 8.XI.2004, Wang XM leg. **Hainan Prov.**: Limushan National Forest Park: 1♂, 21.IV.1996, Peng ZQ leg; 1♂, 10.XI.1989, Ren SX leg; 1♂, Wuzhishan National Forest Park, VIII.1995, Peng ZQ leg. **Yunnan Prov.**: 1♂, Malipo County, 17.VIII.2005, Wang XM leg.

**Distribution.** China: Guangdong, Guangxi, Hainan, Yunnan; Vietnam.

**Remarks.** This species is recorded in China for the first time. The female genitalia of this species are also provided for the first time. 

**Biology.** Unknown.


***Telsimia lunata* Wang sp. nov. 新月寡节瓢虫**


(Figure 7 and Figure 8) 

**Etymology.** The specific epithet is the Latin adjective “*lunata*”, meaning crescent-shaped, referring to the crescent stripe on elytra. 

**Diagnosis.** This species can be easily distinguished from all known *Telsimia* species by its unique elytra markings, as shown in Figure 7. The male genitalia are similar to *T*. *postocula* but can be distinguished by detailed characteristics of tegmen in ventral view: penis guide slightly longer (1.2×) than phallobase and apical notch inverted V-shaped in *T. lunata* (Figure 7j); in *T*. *postocula*, penis guide distinctly longer (1.8×) than phallobase and apical notch nearly fusiform (Kapur 1967, p.169, Figure 7b). 

**Description.** TL: 1.38–1.76 mm, TW: 1.10–1.36 mm, TH: 0.60–0.80 mm, PL: 0.36–0.52 mm, PW: 0.84–0.96 mm, HW: 0.50–0.64 mm, ID: 0.18–0.24 mm, TL/TW: 1.25–1.32, PL/PW: 0.43–0.60, EL/EW: 0.84–1.05, HW/PW: 0.54–0.67, PW/EW: 0.71–0.79, ID/HW: 0.33–0.40. 

Body oval, convex, broadest shortly before middle; elytra black to crater brown, each elytron with two markings as showed in Figure 7a–c; pronotum black to dark brown with anterior corners brownish yellow, anterior and lateral margin dark brown; head dark brown, clypeus golden yellow; underside dark brown, except mouthparts, prothoracic hypomeron, elytral epipleuron, and legs brownish yellow. Head between eyes nearly 0.35 times the width of head; pubescence dense, appressed, and silver-white. Pronotum approximately twice as broad as median length; anterior border arcuate medially, distance between anterior angles 0.6 times that between posterior angles, lateral borders arcuate, posterior and anterior angles briefly rounded; pubescence dense, suberect, and silver-white. Elytra generally shorter than greatest combined breadth; external margins slightly explanate to borders; pubescence dense, suberect, and silver-white. Abdominal postcoxal lines of ventrite 1 not recurved, punctures between coxae small and sparse; ventrite 5 in midline distinctly shorter than ventrites 2–4 together, apical border briefly notched medially in male and evenly arcuate in female (Figure 7d,f). 

Male genitalia: Penis distinctly stout except basal one-third relatively slender, slightly curved at middle part, apex rounded in lateral view (Figure 7g). In ventral view, penis broadest at middle part, apex bifurcated (Figure 7h). Penis guide in lateral view thick, almost parallel at basal two-thirds, then inner side directly narrowing to outside, forming a blunt apex; tegminal strut extremely short (Figure 7i). Penis guide slightly longer than phallobase. Parameres distinctly shorter than penis guide, apex with several long setae. Penis guide in ventral view, 1.60 times as long as wide, nearly parallel for basal two-thirds, apex bifid with a large, inverted V-shaped notch (Figure 7j). 

Female genitalia: As shown in Figure 7e. 

**Types (13). Holotype**: ♂, **China**: **Hainan Prov.**: Diaoluoshan National Nature Reserve, IX.1995, Peng ZQ leg. No. 953187. **Paratypes (12)**: **China**: **Hainan Prov.**: 6♂1♀, Tong Shi City, VIII.1995, Peng ZQ leg; No. 95137–95141 (two specimens without number); 1♂1♀, Le Dong County, Chong Po Town, 10.IV.1996, Peng ZQ leg, No. 963133/963134; 2♀, Qiong Hai County, Dong Ping Farm 13.IV.1996, Peng ZQ leg, No. 953135–953136; 1♀, Bao Ting County, 21.VII.1999, Peng ZQ leg.

**Distribution.** China: Hainan. 

**Biology.** Unknown.


***Telsimia parvus* Wang sp. nov.**
**小叶寡节瓢虫**


(Figure 8 and Figure 9) 

**Etymology.** The specific epithet is the Latin adjective “*parvus*”, meaning small, referring to its small penis guide in ventral view. 

**Diagnosis.** In external appearance, it is similar to *T*. *forcipata* sp. nov., for the short oval body, narrower interocular distance (less than one-third the head width), and dense, gold pubescence on head (Figure 9a–c), but their male genitalia vary greatly (Figure 9e–h and Figure 10g–j). In male genital structure, this species is most closely related to *T. jinyangiensis*, but can be easily distinguished from the latter by some detailed characteristics: in *T. parvus*, basal part of penis slender, without any subsidiary structure (Figure 9e); tegminal strut very short, nearly one-tenth the length of tegmen (Figure 9g); phallobase 1.2 times the length of penis guide; apical emargination of penis guide is small and deep, the depth is five times the width of the opening (Figure 9h); in *T. jinyangiensis*, basal part of penis with a broad handle-like structure; tegminal strut short, nearly one-fifth the length of tegmen; phallobase nearly the same length as penis guide; apical emargination of penis guide is short triangular, the depth is three times the width of the opening. 

**Description.** TL: 1.60 mm, TW: 1.36 mm, TH: 0.92 mm, PL: 0.46 mm, PW: 0.92 mm, HW: 0.56 mm, ID: 0.16 mm, TL/TW: 1.18, PL/PW: 0.50, EL/EW: 0.96, HW/PW: 0.61, PW/EW: 0.68, ID/HW: 0.29. 

Body short oval, convex, broadest at middle; above entirely black; underside dark brown to black, except elytral epipleuron orange-brown, mouthparts and legs brownish yellow. Head between eyes narrow, nearly 0.3 times the width of head; pubescence dense, appressed and gold. Pronotum twice as broad as median length; anterior border arcuate medially, distance between anterior angles 0.6 times that between posterior angles, lateral borders arcuately convergent from obtuse posterior to briefly rounded anterior angles; pubescence dense, suberect and gold. Elytra slightly shorter than greatest combined breadth; external margins not or slightly explanate to borders; pubescence dense, suberect, and silver-white. Abdominal postcoxal lines of ventrite 1 recurved, punctures between coxae small and sparse; ventrite 5 in midline as long as ventrites 2–4 together, apical border distinctly notched medially in male (Figure 9d). 

Male genitalia: Penis robust, in lateral view, basal half gradually narrowing from the middle to distal end, as long as the tegmen; apical half slightly narrowing to a rounded apex (Figure 9e). In lateral view of tegmen (Figure 9g), tegminal strut extremely short, nearly one-tenth the length of tegmen; phallobase 1.2 times the length of penis guide; penis guide thick, thickness almost three-fourth the width of penis guide in ventral view, gradually narrowing from base to apical one-third, then inner side suddenly convergent to a blunt apex; parameres robust at basal two-fifth and slender at apical three-fifth, with several long setae at apex, as long as penis guide. In ventral view of tegmen, the phallobase nearly rectangle, the ratio of width to length is about 3:4; base of penis guide nearly 0.65 times the width of phallobase, penis guide slightly convergent to apical one-third, then gradually narrowing to a bifid apex, the apical emargination small and deep, the depth five times the width of the opening (Figure 9h). 

Female genitalia: Unknown.

**Types. Holotype**: ♂, **China**: **Guangxi Prov.**: Shiwandashan National Forest Park, Hongqi Forest Farm, 11.XI.2004, Zhang CW leg, No. 20051211229, SCAU(E)16742.

**Distribution.** China: Guangxi.

**Biology**. Unknown.


***Telsimia forcipata* Wang sp. nov.**
**钳叶寡节瓢虫**


(Figure 8 and Figure 10) 

**Etymology.** The specific epithet is the Latin adjective “*forcipata*”, meaning forcipate, referring to its forcipate penis guide in ventral view.

**Diagnosis.** Penis guide of this species in ventral view is deeply bifid (Figure 10j), which is similar to *T*. *ismayi* (Chzeau 1984, p. 4, pl. II-4) and *T*. *nigra* (Weise). However, in the last two species, there is a distinct triangular protrusion on each lateral side at apical one-third of penis guide, which can be easily distinguished from *T*. *forcipata*.

**Description.** TL: 1.32–1.50 mm, TW: 1.10–1.28 mm, TH: 0.60–0.84 mm, PL: 0.36–0.48 mm, PW: 0.78–0.92 mm, HW: 0.48–0.54 mm, ID: 0.12–0.14 mm, TL/TW: 1.17–1.26, PL/PW: 0.44–0.52, EL/EW: 0.89–1.05, HW/PW: 0.57–0.63, PW/EW: 0.69–0.73, ID/HW: 0.24–0.29.

Body short oval, strongly convex, broadest at middle; entirely black above; underside dark brown to black, except elytral epipleuron orange-brown, mouthparts and legs brownish yellow. Head between eyes narrow, less than one-third width of head; pubescence dense, appressed, and gold. Pronotum approximately twice as broad as median length; anterior border arcuate medially, distance between anterior angles 0.6 times that between posterior angles, lateral borders arcuately convergent from obtuse posterior to briefly rounded anterior angles; pubescence dense, suberect, and silver-white. Elytra slightly shorter than greatest combined breadth; external margins not or very slightly explanate to borders; pubescence dense, suberect, and silver-white. Abdominal postcoxal lines of ventrite 1 recurved, punctures between coxae large and sparse; ventrite 5 in midline as long as ventrites 2–4 together, apical border flattened medially in male and evenly arcuate in female (Figure 10d,f).

Male genitalia: Penis thick, curved at middle part, apex rounded, as long as the tegmen (Figure 10h). Tegminal strut stout, broad at basal part, as long as phallobase (Figure 10j). Parameres distinctly longer than phallobase and slightly shorter than penis guide, apex with several long setae. Penis guide in lateral view thick, nearly parallel at basal two-thirds, then narrowing to blunt apex (Figure 10g). Penis guide in ventral view forcipate, splitting upward from apex to basal one-fourth (Figure 10j). 

Female genitalia: As shown in Figure 10e. 

**Types. Holotype**: ♂, **China**: **Hainan Prov.**: Diaoluoshan National Nature Reserve, IX.1995, Peng ZQ leg. No. 20051211235. **Paratypes (13)**: **China**: **Hainan Prov.**: 2♂, Le Dong County, Jian Feng Town, 24.IV.2001, Peng ZQ leg (1♂ with No. 20051211236 and SCAU(E)16792); 1♂, Bao Ting County, 21.VII.1999, Peng ZQ leg; 5♂, Hai Kou City, 6.XI.1989, Ren SX leg; 1♂, Hai Kou City, 6.XI.1989, Ou ZJ leg; 1♂, Diaoluoshan National Nature Reserve, 9.V.1996, Peng ZQ leg. No. 953153; 1♂, Dan Zhou City, Chinese Academy of Tropical Agricultural Sciences, 14.IV.1996, Peng ZQ leg. No. 953162; 2♂, Dan Zhou City, Nan Feng Town, VIII.1995, Peng ZQ leg. No. 953175–953176; 1♂, Ding An County, Zhong Rui Farm, 23.IV.1996, Peng ZQ leg. No. 953158. **Guangdong Prov.**: 1♂, Dinghushan National Nature Reserve, 16.V.1989, Ren SX leg. **Guangxi Prov.**: Longzhou County: Pang XF and Pang H leg: 4♂, 27.VII.1985; 11♂2♀, 1.VIII.1985; Nanning City: Pang XF and Pang H leg: 4♂, 3.VIII.1985; 2♂7♀, 4.VIII.1985 (2♀ with No. SCAU(E)16785 and SCAU(E)16780); 1♂7♀, 5.VIII.1985; 1♂2♀, 6.VIII.1985.

**Distribution.** China: Guangdong, Guangxi, Hainan. 

**Biology**. Unknown.


***Telsimia emarginata* Chapin, 1926 整胸寡节瓢虫**


(Figure 8 and Figure 11) 

*Telsimia emarginata* Chapin, 1926: 132 [19]; Liu, 1963: 82 [41]; Pang and Mao, 1979: 100 [33]; Ren et al., 2009: 148 [39]. 

**Diagnosis.** This species can be easily distinguished from other *Telsimia* species by its inflated phallobase (Figure 11h). In ventral view, penis guide nearly half the width of phallobase, nearly 3.5 times of length than width, apex distinctly bifid (Figure 11j). Penis tubular of uniform thickness (Figure 11g).

**Materials examined. China**: **Zhejiang Prov.**: 22 exs, Linhai City, IX.1979, Yang ZX leg. **Fujian Prov.**: 1♂, 30.IV.1979, Huang BK leg. **Guangdong Prov.**: 5♂, Yangshan County, Chengjia Village, 9–14.VII.1997, Tian MY leg (1 specimen with No. SCAU(E)11022 and 20051211231); Huidong City: 1♂, 8.VII.1988, Yu GY leg; 1♂, 11.V.1988, Pang XF leg; 2♂6♀, Guangzhou City, Tianhe District, 20.I.2020, Huo LZ leg (1♀ with No. SCAU(E)16782 and 1♂ with No. SCAU(E)17193). **Guangxi Prov.**: Guilin City: Pang XF leg: 2♂2♀, 23.IX.1987; 1♂, 15.IX.1987; 1♂, IX.1987; 1♂3♀, Mao’er Mountains, Banshanyao, 16.X.2004, Wang XM leg. **Chongqing Prov.**: 1♂, Beipei District, X.1978, Ren SX leg; 3♂1♀, Zhongxian County, 24.VIII.1989, Ren SX leg. **Yunnan Prov.**: 1♂, Hekou County, Xiaoweishan Mountain, 23.IV.2008, Wang XM leg. 

**Distribution.** China: Zhejiang, Fujian, Guangdong, Guangxi, Chongqing, Sichuan, Yunnan; Vietnam. 

**Remarks.** The female genitalia of this species are provided for the first time.

**Biology.** *Aonidiella aurantii* Maskell, *Aulacaspis citri* Chen, *Chrysomphalus aonidum* L., *Parlatoria ziziphi* (Lucas) and *Unaspis yanonensis* (Kuwana) are reported as its prey [12,33,41,42,43]. The related studies on the biological characteristics and predation efficiency of this species have shown that this species has a significant natural inhibitory effect on a variety of shield scales and has broad development and application prospects [44,45,46].

***Telsimia nigra* (Weise, 1879)** 
**黑背寡节瓢虫**


(Figure 8 and Figure 12) 

*Pentilia nigra* Weise, 1879: 149 [47].

*Platynaspis nigra* (Weise, 1900): 422 [48].

*Telsimia nigra*: Chapin, 1926: 131 [19]; Mader, 1955: 814 [49]; Sasaji, 1971: 213 [26]; Yang and Wu, 1972: 126 [27]; Miyatake, 1978: 18 [23]; Yu, 1995: 141 [35].

*Telsimia nigra centralis* Pang and Mao, 1979: 100 [33]; Ren et al., 2009: 150 [39].

**Diagnosis.** This species can be easily distinguished from other *Telsimia* species by its unique male genitalia: penis well developed, apex with a long filiform appendage (Figure 12h); penis guide divided into two lobes at median line in entire length, lateral side of each lobe with a distinct angulation at apical 1/3 (Figure 12i). 

**Materials examined (34). China**: **Jiangxi Prov.**: 1♂, Jinggangshan Mountains, Huangyangjie Peak, 16.VIII.2004, Wang XM leg. **Hunan Prov.**: Yanling County, Shennonggu National Park: 2♂5♀, 1100 m, 7.X.2010, Wang XM et al. leg; 1♂ 2♀, 650 m–800 m, 9.X.2010, Wang XM et al. leg. **Guangdong Prov.**: 2♂, Nanling National Nature Reserve, Xiaohuangshan Mountain, 1.X.2004, Wang XM leg (No. 20050319014 and 20050319026). **Guangxi Prov.**: 2♂, Mao’er Mountains, Jiuniutang, 18.X.2004, Wang XM leg, SCAU(E)17154 and SCAU(E)17155. **Sichuan Prov.**: 1♂2♀, Leibo County, Mamize Nature Reserve, 2600 m, 19–20.IX.2007, Wang XM et al. leg (No. SCAU(E)17156, 13064 and 13066); 2♀, Tianquan County, Labahe Town, 1100 m, 4.X.2007, Wang XM et al. leg (No. SCAU(E)13086 and 13095); 2♂1♀, E’bian County, Heizhugou Town, 1900 m, 22–23.IX.2007, Wang XM et al. leg (2♂ with No. SCAU(E)17158 and 17159); 1♂1♀, Gonggashan National Nature Reserve, Yanzigou scenic area, 1500 m, 28.IX.2007, Wang XM et al. leg (1♂ with No. SCAU(E)10114). **Shaanxi Prov.**: 2♂1♀, Xi’an City, Cuihuashan scenic area, 1300 m, 30.VII.2007, Wang XM et al. leg (2♂ with No. SCAU(E)17157 and 17188). **Taiwan Prov.**: 2♂2♀, Taichung City, Heping District, Anma Shan, 25.VII.1971, Yang C. T. leg (1♂ with No. SCAU(E)16786, 1♀ with No. SCAU(E)17217); Jiayi County, Fenchihu Lake, 1400 m, J. Klapperich leg: 1♂, 10.IV.1977, SCAU(E)11026; 1♂, 19.V.1977. 

**Distribution.** China: Fujian, Jiangxi, Henan, Hunan, Guangdong, Guangxi, Sichuan, Yunnan, Shaanxi, Taiwan; Japan. 

**Remarks.** Pang and Mao [33] described a new subspecies according to one specimen from Sichuan Province and explained its difference from that of Japan by the angulation on the lateral side of each lobe and the depth of the split. We compared 11 male specimens from five provinces (including Sichuan Province, the type locality of this subspecies) (Appendix A), and we did not find any significant differences from Sasaji’s illustrations [26] (p.214, Figure 88). 

**Biology.** This species is reported to prey on *Aulacaspis rosarum* Borchsenius, *Unaspis yanonensis* (Kuwana), *Aphis citricidus* (Kirkaldy), and *Lepidosaphes tubulorum* Ferris [7,50,51,52].


***Telsimia darjeelingensis* Kapur, 1969 大吉岭寡节瓢虫**


(Figure 13 and Figure 14) 

*Telsimia darjeelingensis* Kapur, 1969: 50 [25].

**Diagnosis.** Male genitalia of this species resemble *T*. *ceylonica* Kapur 1969, but can be easily distinguished from the latter by the following characteristics: in *T. darjeelingensis*, tegminal strut slightly longer than penis guide; penis guide gradually narrowing to a bifid apex (Figure 13g); basal part of penis with membranous appendages, preapical part of penis slightly enlarged but the apex curved and acute (Figure 13e); in *T*. *ceylonica*, tegminal strut distinctly shorter than penis guide; penis guide gradually narrowing to a blunt apex; basal part of penis without membranous appendages, apex of penis bulbous, nearly dumbbell-shaped (Kapur 1969, p.47, Figure 1).

**Material examined (1).** 1♂, **China**: **Yunnan Prov.**, Gaoligong Mountains, 19.IX.2006, Wang XM leg., SCAU(E)16771.

**Distribution.** China: Yunnan; India.

**Remarks.** This species is recorded in China for the first time.

**Biology**. Unknown.


***Telsimia chayuensis* Wang sp. nov.**
**察隅寡节瓢虫**


(Figure 14 and Figure 15) 

**Etymology.** The specific name refers to its type locality. 

**Diagnosis.** In male genital structure, this species is closely related to *T. emarginata* and *T*. *abdita* Ślipiński, Pang and Pope, 2005, but can be distinguished from the latter by some detailed characteristics: basal one-fourth of the penis thread-like, apical one-fourth gradually narrowing to a blunt apex (Figure 15i); in the latter two species, penis tubular, thickness almost consistent in the whole length, except the apical part slightly narrowing to a rounded apex (Figure 11g). Tegmen is mostly similar to *T*. *abdita*, but can be distinguished from the latter by detailed characters of penis guide: in lateral view, the apex bent inward (Figure 15h), while, in *T*. *abdita*, the apex bent outward; in ventral view, the apex is truncate, with a median line (Figure 15i), while, in *T*. *abdita*, the apex is rounded, without a median line. 

**Description.** TL: 1.70–2.00 mm, TW: 1.28–1.50 mm, TH: 0.76–0.90 mm, PL: 0.40–0.52 mm, PW: 0.90–1.10 mm, HW: 0.56–0.64 mm, ID: 0.24–0.30 mm, TL/TW: 1.31–1.33, PL/PW: 0.43–0.53, EL/EW: 1.00–1.08, HW/PW: 0.58–0.62, PW/EW: 0.67–0.79, ID/HW: 0.41–0.47.

Body oblong oval, convex, broadest at middle; elytra entirely black to dark brown; pronotum black except lateral margin and anterior corners orange-brown, head and clypeus orange-brown; underside dark brown except mouthparts, prothoracic hypomeron, elytral epipleuron, and legs brownish yellow. Head between eyes nearly 0.45 times the width of head; pubescence dense, suberect, and silver-white. Pronotum approximately twice as broad as median length; anterior border arcuate medially, distance between anterior angles 0.60 times that between posterior angles, lateral borders arcuately convergent from obtuse posterior to briefly rounded anterior angles; pubescence dense, suberect, and silver-white. Elytra slightly longer than greatest combined breadth; external margins not explanate to borders; pubescence dense, suberect, and silver-white. Abdominal postcoxal lines of ventrite 1 recurved, punctures between coxae large and dense; ventrite 5 in midline distinctly shorter than ventrites 2–4 together, apical border briefly notched medially in male and evenly arcuate in female (Figure 15d,f). 

Male genitalia: Penis, in lateral view (Figure 15g), slightly shorter than tegmen, basal one-fourth thread-like, apical one-fourth gradually narrowing to a blunt apex. In lateral view of tegmen (Figure 15h), tegminal strut short, nearly one-sixth the length of tegmen; phallobase inflated; penis guide slender, gradually narrowing to a blunt apex, which bent inward; parameres slender, nearly the same length as penis guide. In ventral view of tegmen (Figure 15i), the phallobase nearly rectangle, the ratio of width to length is about 3:2; penis guide nearly parallel from base to apical one-third, then gradually narrowing to a truncate and protruding apex, apical one-fourth of penis guide has a midline. 

Female genitalia: As shown in Figure 15e. 

**Types. Holotype**: ♂, **China**: **Tibet**: Chayu County, Cibagou National Nature Reserve, ca. 2000 m, 13.X.2007, Wang XM et al. leg, SCAU(E)16767. **Paratypes (6)**: 1♂5♀, with same data as holotype except one female with date 17.X.2007. 

**Distribution.** China: Tibet.

**Biology.** Unknown.


***Telsimia lobatus* Wang sp. nov.**
**浅裂寡节瓢虫**


(Figure 14 and Figure 16) 

**Etymology.** The specific epithet is the Latin adjective “*lobatus*”, meaning lobed, referring to the bifid apex of penis guide in ventral view. 

**Diagnosis.** This species can be easily distinguished from all known *Telsimia* species by its male genitalia: penis distinctly stout (Figure 16f), penis guide in ventral view broad, 2.5 times of length than width, parallel at basal two-thirds, then gradually narrowing to a very small bifid apex (Figure 16h). 

**Description.** TL: 1.30–1.50 mm, TW: 1.10–1.20 mm, TH: 0.70–0.84 mm, PL: 0.38–0.46 mm, PW: 0.80–0.90 mm, HW: 0.50–0.56 mm, ID: 0.20–0.24 mm, TL/TW: 1.18–1.26, PL/PW: 0.44–0.53, EL/EW: 0.86–1.03, HW/PW: 0.56–0.65, PW/EW: 0.72–0.78, ID/HW: 0.37–0.44. 

Body short oval, convex, broadest at middle; above black except apical part narrowly brownish; underside dark brown to black, except mouthpart and legs orange-brown. Head between eyes nearly 0.4 times the width of head; pubescence dense, appressed, and silver-white. Pronotum approximately twice as broad as median length; anterior border arcuate medially, distance between anterior angles 0.6 times that between posterior angles, lateral borders arcuate, posterior and anterior angles briefly rounded; pubescence dense, suberect, and silver-white. Elytra generally shorter than greatest combined breadth; external margins not explanate to borders; pubescence dense, suberect, and silver-white. Abdominal postcoxal lines of ventrite 1 distinctly recurved, punctures between coxae large and dense; ventrite 5 in midline as long as or slightly shorter than ventrites 2–4 together, apical border nearly flattened medially in male (Figure 16d). 

Male genitalia: Penis, in lateral view (Figure 16f), distinctly stout, curved at middle part, apex rounded; in ventral view of tegmen (Figure 16h), apical half gradually narrow to apex. Tegminal strut nearly one-third the length of main part of the tegmen. Parameres slightly shorter than penis guide, apical part with several long setae. Penis guide thick in lateral view (Figure 16g), gradually narrowing to a pointed apex; in ventral view broad, 2.5 times of length than width, parallel at basal two-thirds, and then gradually narrowing to a small bifid apex. 

Female genitalia: Unknown. 

**Types. Holotype**: ♂, **China**: **Hainan Prov.**: Diaoluoshan National Nature Reserve, IX.1995, Peng ZQ leg. No. 953188. **Paratypes (6)**: **China**: **Hainan Prov.**: 1♂, Diaoluoshan National Nature Reserve, 26.VII.2006, Wang XM leg, SCAU(E)14534; 1♂, Bawangling National Nature Reserve, 5.V.2005, Wang XM leg, No. 20070320021, SCAU(E)16740; 2♂, Jianfengling National Nature Reserve, 11.XI.1989, Ren SX leg; 1♂, Jianfengling National Nature Reserve, 12.XI.1989, Ren SX leg; 1♂, Jianfengling National Nature Reserve, 13.XI.1989, Ou ZJ leg. 

**Distribution.** China: Hainan. 

**Biology.** Unknown.


***Telsimia humidiphila* Kapur, 1969**
**短叶寡节瓢虫**


(Figure 14 and Figure 17) 

*Telsimia humidiphila* Kapur, 1969: 52 [25].

**Diagnosis.** Male genitalia of this species are similar to *T*. *postocula* but can be easily distinguished from the latter by its slenderer penis guide and shorter parameres. In *T*. *humidiphila*, penis guide 2.5–3 times the length than width, parameres half the length of penis guide (Figure 17j); in *T*. *postocula*, penis guide nearly 2 times the length than width, parameres slightly shorter than penis guide (Kapur 1967, p.169, Figure 7b). 

**Materials examined (15). China**: **Guangdong Prov.**: 2♂, Yangshan County, Chengjia Village, 9–14.VII.1997, Tian MY leg (1♂ with No. 20051211234); **Guangxi Prov.**: Mao’er Mountains: Wang XM leg: 1♂, Shili Canyon, 19.X.2004; 1♂1♀, Banshanyao, 19.X.2004, No. 20061221095 and 20061222020; 1♂1♀, Gaozhai, 20.X.2004, No. 20061218026; Shiwandashan National Forest Park: 1♂, Hongqi Forest Farm, 9.XI.2004, Lv XB leg; 1♂, Fulong Mountain, 7.XI.2004, Wang XM leg, No. 20050404012. 1♂, 9–11.XI.2004, Wang XM leg, SCAU(E)14004. **Guizhou Prov.**: 1♂, Leigongshan Mountains, Xiaodanjiang, 13.VIII.2006, Wang XM leg, SCAU(E)13315. **Hainan Prov.**: 1♂, Danzhou City, Nada Town, 8.XI.1989, Ren SX leg; 1♂, Tongshi City, VIII.1995, Peng ZQ leg, No. 953221; 2♂, Wenchang City, Dongjiao Town, IX.1995, Peng ZQ leg, No. 953222 and 953223. 

**Distribution.** China: Guangdong, Guangxi, Hainan, Guizhou; India.

**Biology.** This species has been reported as preying on diaspidine scales in coconut plantations on Hainan Island [37], representing the first report of this species in China. 


***Telsimia nagasakiensis* Miyatake, 1978**
**长崎寡节瓢虫**


(Figure 14 and Figure 18) 

*Telsimia nagasakiensis* Miyatake, 1978: 13 [23]; Park and Yoon, 1993: 277 [53]; Yu, 1995: 141 [35]; Yu and Lau, 2001: 163 [54]; Ren et al., 2009: 150 [39]. 

**Diagnosis.** See that of *T. chujoi*.

**Materials examined (25). Holotype:** [Kyushu], Nagasaki City, 4.IV.1973, Y. Furuki// Holotype, *Telsimia nagasakiensis* Miyatake//1103 (♂, EUMJ). **Other materials: China**: **Taiwan Prov.**: 2♂3♀, Tainan County, Wushantou Dam, 20.VI.1971, Yang C. T. leg; 1♂1♀ and 12 exs., Taichung City, 24.II.1972, Yang C. T. leg; 1 ex., Liukuei Kaohsiung Hsien, 27–28.VI.1981, L. Y. Chou and C. H. Yang leg; 2♂and 3 exs., Ouluanpi Coast Forest, Pingtung Hsien, 7-9.XII.1982, S. C. Lin and S. P. Huang leg.

**Distribution.** China: Hong Kong, Taiwan; Japan; Korea. 

**Remarks.** The female genitalia of this species are provided for the first time (Figure 18e).

**Biology.** Detailed larval description and additional adult diagnosis have been reported by Park and Yoon [53]. Some biological notes, including habitats, prey, seasonal observation, and the distributional tendency of the species, are also discussed.


***Telsimia chujoi* Miyatake, 1959**
**朱氏寡节瓢虫**


(Figure 19 and Figure 20) 

*Telsimia chujoi* Miyatake, 1959: 46 [21]; Nakane, 1959: 49 [55]; Miyatake, 1978: 17 [23]; Yang and Wu, 1972: 126 [27].

**Diagnosis.** Male genitalia of this species are similar to *T*. *nagasakiensis* Miyatake, 1978, but can be distinguished by the combined characteristics as follows: in *T*. *chujoi*, tegminal strut slightly shorter than penis guide, penis guide gradually narrowing to a pointed apex, apex with a short median line (Figure 19g); in *T*. *nagasakiensis*, tegminal strut nearly half the length of the penis guide, the penis guide gradually narrowing to a blunt apex, apex with a narrow, short median emargination (Figure 18i).

**Material examined (1).** 1♂, **China**: **Taiwan Prov.**, Pingdong County, Nanren Mountain, 3.IV.2002, Chen WH leg, SCAU(E)16776. 

**Distribution.** China: Taiwan; Japan.

**Biology.** This species is reported to be the natural enemy of the lac insect *Kerria lacca* (Kerr.) in Taiwan [56]. 


**Telsimia sichuanensis Pang and Mao, 1979 四川寡节瓢虫**


(Figure 20 and Figure 21) 

*Telsimia sichuanensis* Pang and Mao, 1979: 101 [33]; Ren et al., 2009: 152 [39]. 

**Diagnosis.** This species can be easily distinguished from other *Telsimia* species by its unique male genitalia: in lateral view of tegmen (Figure 21g), tegminal strut very short, nearly one-sixth the length of the tegmen, main tegmen slender, gradually narrowing from base to apex, penis guide exceptionally long, more than twice the length of parameres; in ventral view (Figure 21i), penis guide six times the length as width, nearly parallel from base to apical one-sixth, then gradually narrowing to a blunt apex. Penis thread-like, distinctly longer than the penis guide (Figure 21h).

**Materials examined (21). China: Zhejiang Prov.**: 1♂, Cixi City, 30.VII.1988, Yu GY leg. **Fujian Prov.**: Shaowu City: 1♂ and 5 exs., 24.VIII.1984, Pang XF leg; 1♂1♀, 16.IV.1981, Huang BK leg; 1♀, 6.XI.1982, Huang BK leg. **Jiangxi Prov.**: 1♂, Taihe County, 12.VIII.2004, Wang XM leg, No. 20051211230, SCAU(E)11021. **Hubei Prov.**: 1♂1♀, Yichang City, 1978, Ying RS leg. **Chongqing Prov.**: Zhong County: 1♂, 23.VIII.1989, Ren SX leg; 3♂, 24.VIII.1989, Ren SX leg; 1♂, Wuxi County, 2.VII.1989, Ren SX leg. **Guangxi Prov.**: 1♂, Longzhou County, 1.VIII.1985, Pang H leg. **Hainan Prov.**: 1♂, Haikou City, 3.XI.1989, Ren SX leg; 1♂, Qiongzhong County, Yangjiang Farm, IX.1995, Peng ZQ leg. No. 953189. 

**Distribution.** China: Zhejiang, Fujian, Jiangxi, Hubei, Guangxi, Hainan, Sichuan, Shaanxi, Hong Kong.

**Biology.** This species is reported to prey on *Hemiberlesia pitysophila* Takagi in Guangxi, China [57].


***Telsimia shirozui* Miyatake, 1965**
**台湾寡节瓢虫**


(Figure 20 and Figure 22) 

*Telsimia shirozui* Miyatake, 1965: 55 [22]; Miyatake, 1978: 18 [23].

**Diagnosis.** Male genitalia of this species are closely related to *T*. *nitida* Chapin 1926, but can be easily distinguished from the latter by the ventral view of tegmen: penis guide slightly narrower than phallobase, apical emargination small, width of notch end nearly seven times of width of penis guide at middle part (Figure 22i); in *T*. *nitida*, penis guide distinctly narrower than phallobase, apical emargination larger, width of notch end nearly twice the width of penis guide at middle part (Chapin 1965, p. 242, Figure 39). 

**Materials examined (22). Holotype:** “[Formosa], Jitsugetsu-tan, Jun.26.1961, T. Shirozu leg. //Holotype, *Telsimia shirozui* Miyatake//1182” (♂, EUMJ). **Other materials: China**: **Taiwan Prov.**: 1♂1♀, Tainan County, Wushantou Dam, 20.VI.1971, Yang C. T. leg; 5♂2♀ and 12 exs., Taichung City, 24.II.1972, YANG C. T. leg.

**Distribution.** China: Taiwan. 

**Remarks.** The female genitalia of this species are provided for the first time (Figure 22e).

**Biology.** Unknown.


***Telsimia jinyangiensis* Pang and Mao, 1979 金阳寡节瓢虫**


(Figure 20 and Figure 23) 

*Telsimia jinyangiensis* Pang and Mao, 1979: 102 [33]; Ren et al., 2009: 150 [39].

**Diagnosis.** Male genitalia of this species are similar to *T*. *elainae* Chazeau 1984, especially the lateral view of penis and tegmen, but can be easily distinguished from latter by the ventral view of tegmen: in *T*. *jinyangiensis*, penis guide as long as phallobase, and distinctly narrower than phallobase, apical emargination small and inverted V shape (Figure 23i); in *T*. *elainae*, penis guide distinctly longer than phallobase, and as wide as phallobase, apical emargination larger and inverted U shape (Chazeau 1984, p. 2, pl. I-4). 

**Material examined. China**: **Sichuan Prov.**: 1♂3♀, Puge County, 15.IX.2005, Wang XM et al. leg. (1♂ with No. 20071201032, 2♀ with No. SCAU(E)13752 and SCAU(E)11000).

**Distribution.** China: Guangxi, Sichuan, Yunnan.

**Remarks.** The female genitalia of this species are provided for the first time (Figure 23e).

**Biology.** Unknown.


***Telsimia huiliensis* Pang and Mao, 1979**
**会理寡节瓢虫**


(Figure 20 and Figure 24) 

*Telsimia huiliensis* Pang and Mao, 1979: 102 [33]; Ren et al., 2009: 148 [39].

**Diagnosis.** Male genitalia of this species resemble *T. kuznetsovi* Hoàng 1987 but can be easily distinguished from the latter by the following characteristics: penis guide thinner and longer, upward crack of penis guide opening relatively longer and narrower (Figure 24h). 

**Material examined.** 1♂, **China**: **Sichuan Prov.**: Miyi County, 30.IX.2000, Peng ZQ leg, SCAU(E)16784.

**Distribution.** China: Sichuan, Yunnan.

**Biology.** Unknown. 

### 3.2. Key to Chinese Species of Telsimia

1Dorsum not entirely black, partially with yellow spots……………………………………………… **2**

-Dorsum entirely black……………………………………………………………………………………………… **7**

2Pronotum entirely yellow………………………………………………………………………………………… **3**

-Pronotum black with at least lateral margins or anterior corners yellowish brown **5**

3Each elytrum with a large yellow spot at terminal half (Figure 1a–c); male genitalia as in Figure 1g–i……………………………………………… ***scymnoides* Miyatake**

-Each elytrum with a small apical spot (Figure 3a–c and Figure 4a–c)……………………………………………… **4**

4Abdominal postcoxal lines of ventrite 1 not recurved (Figure 3d); male genitalia as in Figure 3e–g……………………………………………… ***parascymnoides* sp. nov.**

-Abdominal postcoxal lines of ventrite 1 distinctly recurved (Figure 4d,f); male genitalia as in Figure 4g–i……………………………………………… ***menglaensis* sp. nov.**

6Elytra entirely black; male genitalia as in Figure 5e–g……………………………………………… ***latus* sp. nov.**

-Elytra black with yellow spots……………………………………………… **6**

6Elytra black, each with a small apical spot (Figure 6a–c); male genitalia as in Figure 6g–i……………………………………………… ***elongate* Hoàng**

-Elytra black, each with two brownish yellow markings as in Figure 7a–c; male genitalia as in Figure 7g–j ……………………………………………… ***lunata* sp. nov.**

7Interocular distance less than 1/3 width of head ……………………………………………… **8**

-Interocular distance more than 1/3 width of head ……………………………………………… **9**

8Pubescence on pronotum golden (Figure 9a); male genitalia as in Figure 9e–h ……………………………………………… ***parvus* sp. nov.**

-Pubescence on pronotum silver-white (Figure 10a); male genitalia as in Figure 9g–j ……………………………………………… ***forcipata* sp. nov.**

9Interocular distance more than 1/2 width of head ……………………………………………… **10**

-Interocular distance 1/3-1/2 width of head ……………………………………………… **11**

10Abdominal postcoxal lines of ventrite 1 not recurved, apical border of ventrite 5 distinctly notched medially in male (Figure 11f); male genitalia as in Figure 11g–j ……………………………………………… ***emarginata* Chapin**

-Abdominal postcoxal lines of ventrite 1 distinctly recurved, apical border of ventrite 5 evenly rounded in male (Figure 12f); male genitalia as in Figure 12g–i ……………………………………………… ***nigra* (Weise)**

11Body oblong oval (1.30–1.35: 1)……………………………………………… **12**

-Body short oval (1.15–1.28: 1)……………………………………………… **13**

12Pubescence on frons dense; pubescence on pronotum and elytra sub-depressed, variably directed (Figure 13b); male genitalia as in Figure 13e–g……………………………………………… ***darjeelingensis* Kapur**

-Pubescence on frons sparse; pubescence on pronotum and elytra sub-depressed, directed backward (Figure 15b); male genitalia as in Figure 15g–i ……………………………………………… ***chayuensis* sp. nov.**

13Dorsum strongly convex, elytral height more than 0.5 times of elytral length……………………………………………… **14**

-Dorsum normally convex, elytral height less than 0.45 times of elytral length……………………………………………… **16**

14Smaller species (1.30–1.50 mm long); male genitalia as in Figure 16e–h ……………………………………………… ***lobatus* sp. nov.**

-Larger species (1.50–1.90 mm long)……………………………………………… **15**

15Pubescence on frons short, directed towards midline (Figure 17a); clypeus relatively longer, with a tuft of “beard” at middle, directed downward; male genitalia as in Figure 17g–j……………………………………………… ***humidiphila* Kapur**

-Pubescence on frons longer, directed downward (Figure 18a); clypeus relatively shorter, without that “beard”; male genitalia as in Figure 18g–i………………………………………………***nagasakiensis*** **Miyatake**

16Abdominal postcoxal lines of ventrite 1 not recurved……………………………………………… **17**

-Abdominal postcoxal lines of ventrite 1 distinctly recurved ……………………………………………… **18**

17Pubescence on elytra variably directed (Figure 19a–c); punctures on abdomen smaller (Figure 19d); distributed in Taiwan and Japan; male genitalia as in Figure 19e–g……………………………………………… ***chujoi* Miyatake**

-Pubescence on elytra directed backward (Figure 21a–c); punctures on abdomen larger, especially that on ventrite 1–2 (Figure 21f); distributed in mainland China; male genitalia as in Figure 21g–i……………………………………………… ***sichuanensis* Pang and Mao**

18Punctures on abdomen smaller (Figure 22d,f); distributed in Taiwan; male genitalia as in Figure 22g–i……………………………………………… ***shirozui* Miyatake**

-Punctures on abdomen larger, especially that on ventrite 1 (Figure 23d and Figure 24d); distributed in Sichuan ……………………………………………… **19**

19Interocular distance narrower, nearly 1/3 width of head (Figure 23a); lateral margins of pronotum nearly straight; male genitalia as in Figure 23g–i ……………………………………………… ***jinyangiensis* Pang and Mao**

-Interocular distance broader, nearly 1/2 width of head (Figure 24a); lateral margins of pronotum arched; male genitalia as in Figure 24e–h ……………………………………………… ***huiliensis* Pang and Mao**

## 4. Discussion

**Species identification.** Elytral markings are often used as one of the main distinguishing features of Coccinellidae, but this is only suitable for a few species in this group, such as *T*. *lunata* sp. nov., *T*. *flavomaculata* Poorani 2003, *T*. *kuznetsovi* Hoàng 1987, *T*. *scymnoides* Miyatake 1978, *T*. *bicolor* Kapur 1969, and *T*. *postocula* Kapur 1967, while most the other species are all black or with a little yellow spot at the end of the elytra, which hinders identification by elytral markings. In addition, color variation is common in many groups of Coccinellidae, although it has not been found in this group; its potential impact on species identification should be fully considered. Moreover, body size, shape, and punctures are occasionally used for species identification. However, as the number of species and specimens increases, the differences between species become smaller and individual differences become greater, making species identification using these characteristics very difficult. In the future, with the extensive use of 3D scanning electron microscopy and micro-CT, combined with new techniques such as geometric morphology, computer vision, and deep learning [58,59], it may be possible to make new breakthroughs in the direction of external morphology-based species identification. 

Some scholars have also used female genitalia as a species identification feature, such as with *T*. *tamdaoensis* Hoàng 1981 and *T*. *bangalorensis* Kapur 1969. However, female genitalia are considered more applicable to distinguishing different species rather than identifying new species. The reason is that if female genitalia were to be used for species identification, strictly speaking, a comparative study of the female genitalia of all previous species would be necessary. The reality is that the female genitalia of many species have not been previously studied, and many species have been described on the basis of males only. In addition, male–female matching, a major problem in taxonomic studies, is also present in this taxon. In the present study, we matched males and females on the basis of a large number of external morphological characteristics and the same label information. However, this requires special caution, as multiple species could be collected at the same time and place. Through our study, we found that female genitalia differed little between most species in this taxon. Therefore, the use of female genitalia as a species identification feature in this taxon is not recommended. 

In conclusion, male genitalia are more recommended for species identification of this taxon, and other characteristics can be used as an auxiliary. It is also possible to try to use DNA barcoding for species identification and male–female matching; this is one of the directions of future research. 

**Phylogeny.** So far, we have not seen any phylogenetic studies within the genus *Telsimia*. Through this study, we found that there were differences in the ratio of male genitalia to body size among different groups. For example, in *T*. *sichuanensis*, *T*. *chujoi*, *T*. *nagasakiensis*, *T*. *elongate*, and *T*. *darjeelingensis*, the penis is obviously longer than the abdomen; in *T*. *nigra*, the main part of the penis is about equal to the abdomen; in *T*. *scymnoides* and *T*. *parascymnoides* sp. nov., the penis is slightly shorter than the abdomen; in the other groups, the penis is equal to or slightly longer than half the length of the abdomen. This attribute may be sourced from reproductive competition, and the changing trend of the ratio may indicate the direction of the species evolution of this taxon, forming an important feature of higher-level classification. Therefore, in the illustrations of ladybug taxonomic articles, we strongly recommend the use of scale bars. When the difference in genital size between species is not particularly large, comparable images with a particular scale bar will make the ratio difference more intuitive. In addition, antennae, mouthparts, feet, and prothorax are also important classification features for phylogenetic studies, and further research will be carried out in this area in the future. 

**Biology and biological control.** There are relatively few studies on the biology of species in this taxon [7,43,44,45,53], and the biology of most species is unknown. Research on biological control applications is also limited to a few species, such as *T. emarginata*, *T*. *nigra*, and *T*. *nitida* [4,5,9,12]. In addition to the widespread distribution of these species, the reason for this situation is largely due to the difficulty of species identification. Therefore, it is hoped that our study may provide some help. 

## Figures and Tables

**Figure 1 insects-13-00869-f001:**
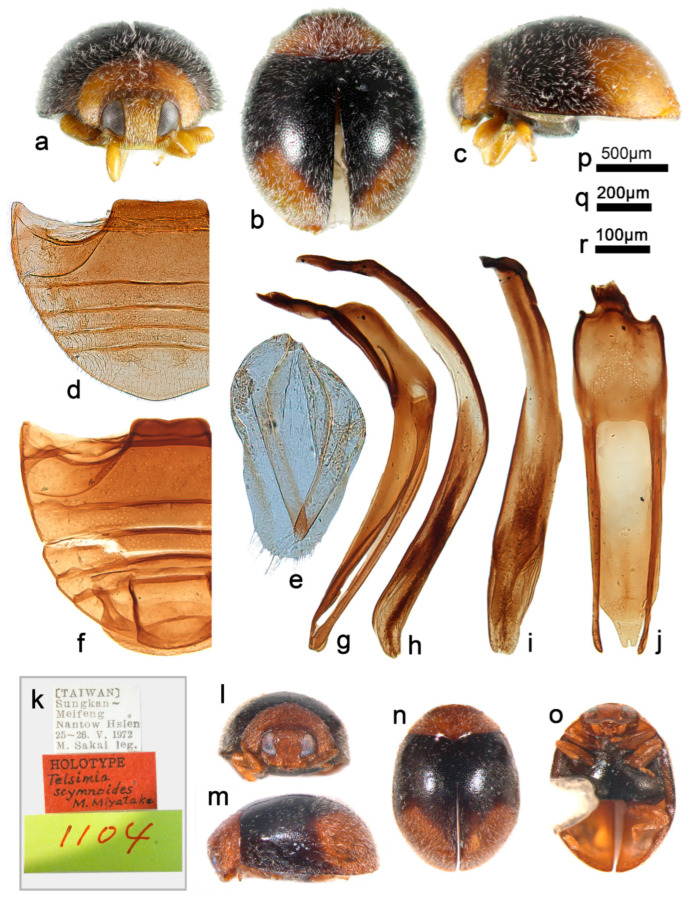
*Telsimia scymnoides* Miyatake. (**a**–**c**) Adults: frontal, dorsal, and lateral view; (**d**) Abdomen, female; (**e**) ovipositor; (**f**) abdomen, male; (**g**) penis guide, lateral view; (**h**) penis, lateral view; (**i**) penis, ventral view; (**j**) penis guide, ventral view; (**k**–**o**) Holotype**. Scale bars:** a, b, c—(**p**), abdomen—(**q**), bisexual genitalia—(**r**).

**Figure 2 insects-13-00869-f002:**
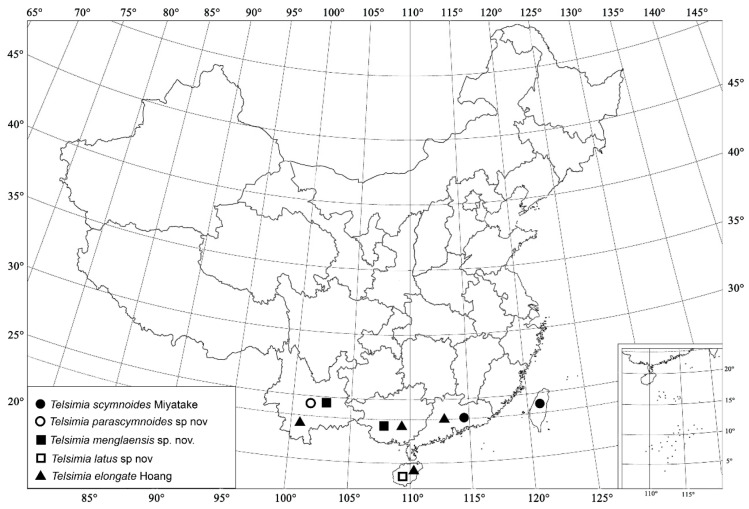
Distribution map. (●) *Telsimia scymnoides* Miyatake; (○) *Telsimia parascymnoides* sp. nov.; (■) *Telsimia menglaensis* sp. nov.; (□) *Telsimia latus* sp. nov.; (▲) *Telsimia elongate* Hoàng.

**Figure 3 insects-13-00869-f003:**
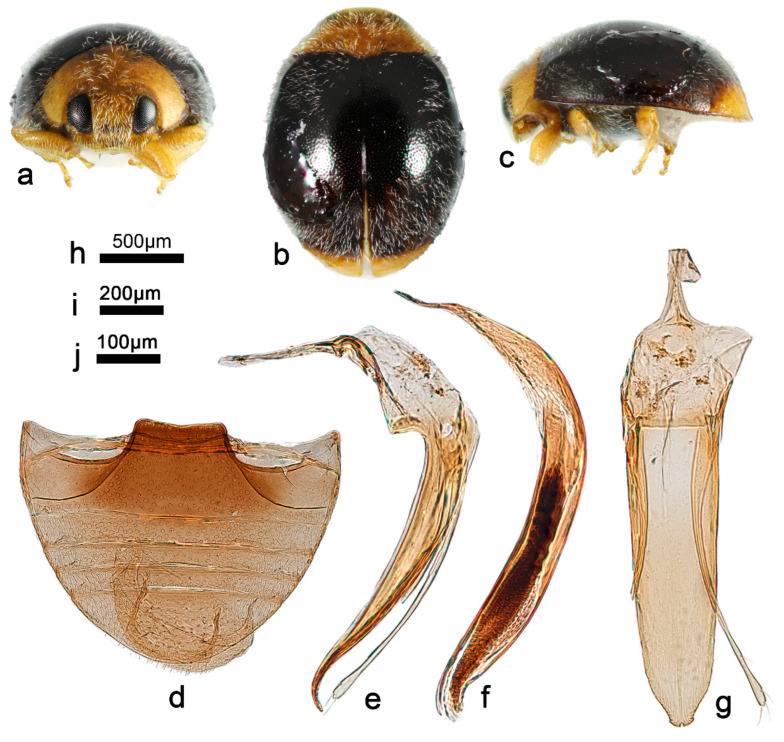
*Telsimia parascymnoides* sp. nov. (**a–c**) Adult: frontal, dorsal, and lateral view; (**d**) Abdomen, male; (**e**) penis guide, lateral view; (**f**) penis, lateral view; (**g**) penis guide, ventral view. **Scale bars:** Adult—(**h**), abdomen—(**i**), male genitalia—(**j**).

**Figure 4 insects-13-00869-f004:**
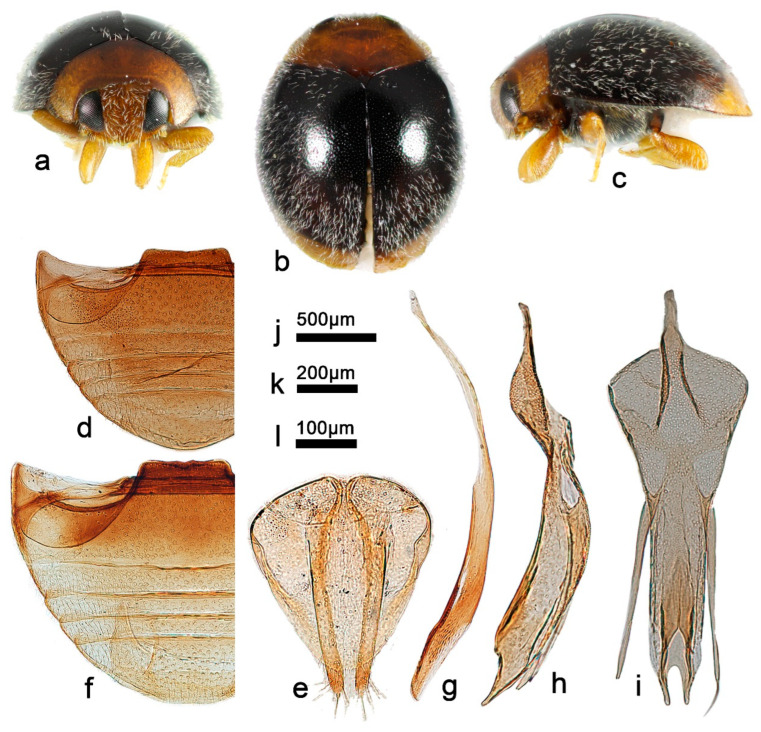
*Telsimia menglaensis* sp. nov. (**a**–**c**) Adult: frontal, dorsal, and lateral view; (**d**) Abdomen, female; (**e**) ovipositor; (**f**) abdomen, male; (**g**) penis, lateral view; (**h**) penis guide, lateral view; (**i**) penis guide, ventral view. **Scale bars:** Adult—(**j**), abdomen—(**k**), bisexual genitalia—(**l**).

**Figure 5 insects-13-00869-f005:**
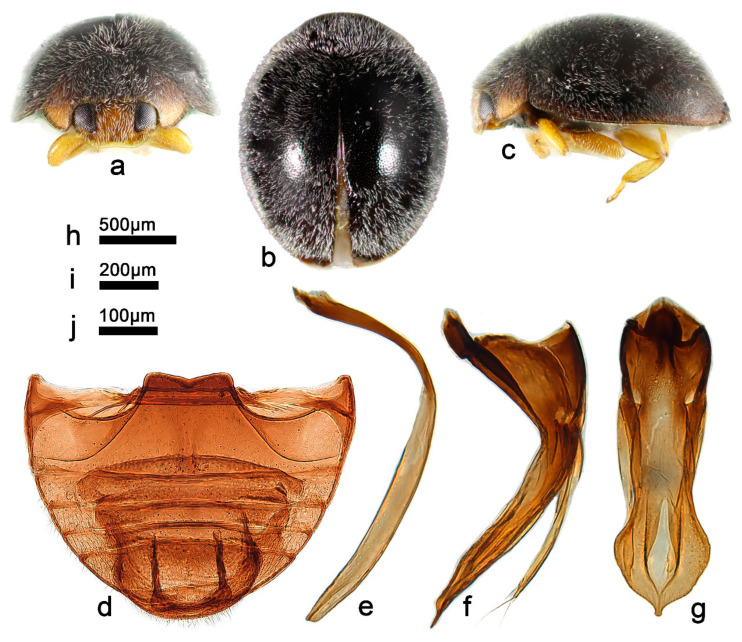
***Telsimia latus* sp.** nov. (**a–c**) Adult: frontal, dorsal, and lateral view; (**d**) Abdomen, male; (**e**) penis, lateral view; (**f**) penis guide, lateral view**;** (**g**) penis guide, ventral view. **Scale bars:** Adult—(**h**), abdomen—(**i**), male genitalia—(**j**).

**Figure 6 insects-13-00869-f006:**
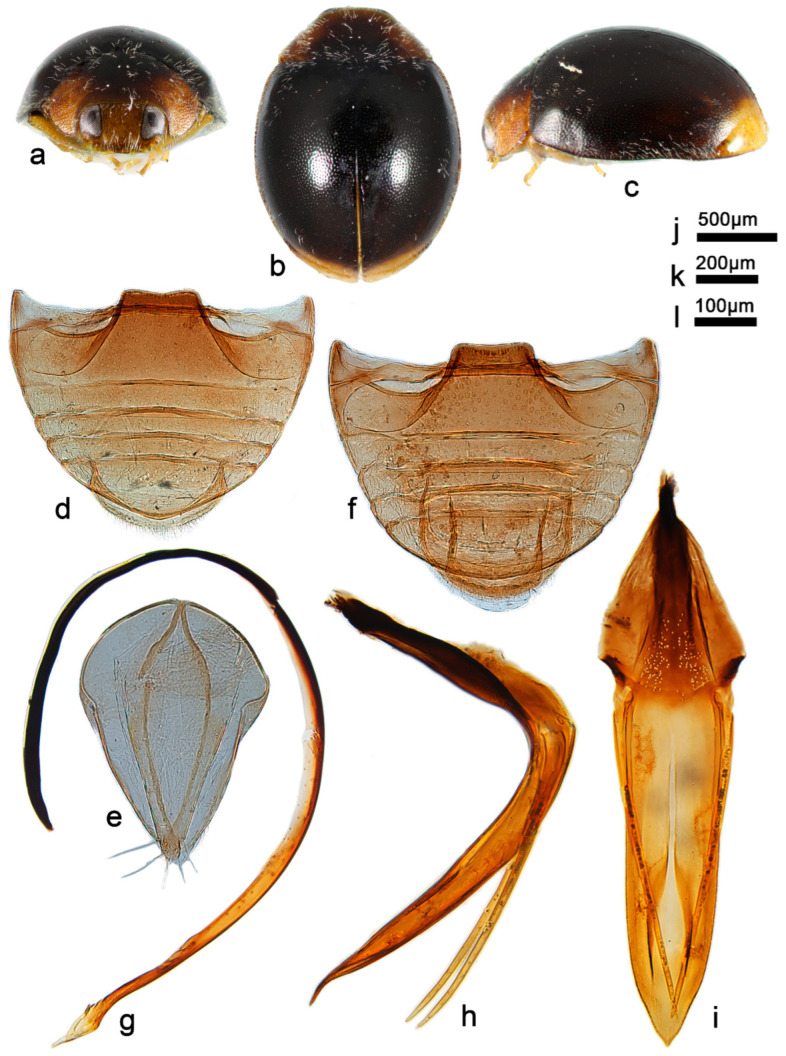
*Telsimia elongate* Hoàng. (**a**–**c**) Adult: frontal, dorsal, and lateral view; (**d**) Abdomen, female; (**e**) ovipositor; (**f**) abdomen, male; (**g**) penis, lateral view; (**h**) penis guide, lateral view; (**i**) penis guide, ventral view. **Scale bars:** Adult—(**j**), abdomen—(**k**), bisexual genitalia—(**l**).

**Figure 7 insects-13-00869-f007:**
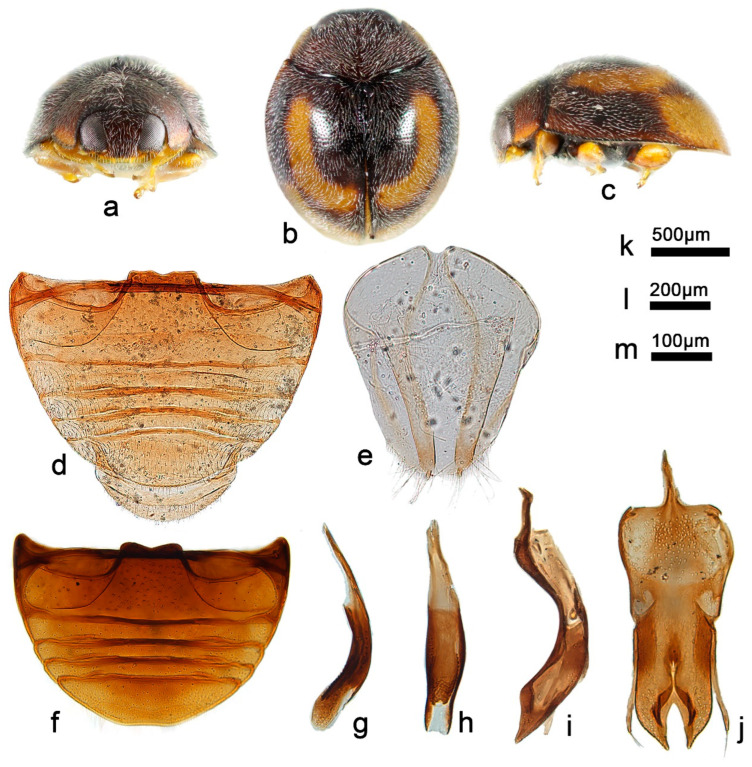
*Telsimia lunata* sp. nov. (**a**–**c**) Adult: frontal, dorsal, and lateral view; (**d**) Abdomen, female; (**e**) ovipositor; (**f**) abdomen, male; (**g**) penis, lateral view; (**h**) penis, ventral view; (**i**) penis guide, lateral view; (**j**) penis guide, ventral view. **Scale bars:** Adult—(**k**), abdomen—(**l**), bisexual genitalia—(**m**).

**Figure 8 insects-13-00869-f008:**
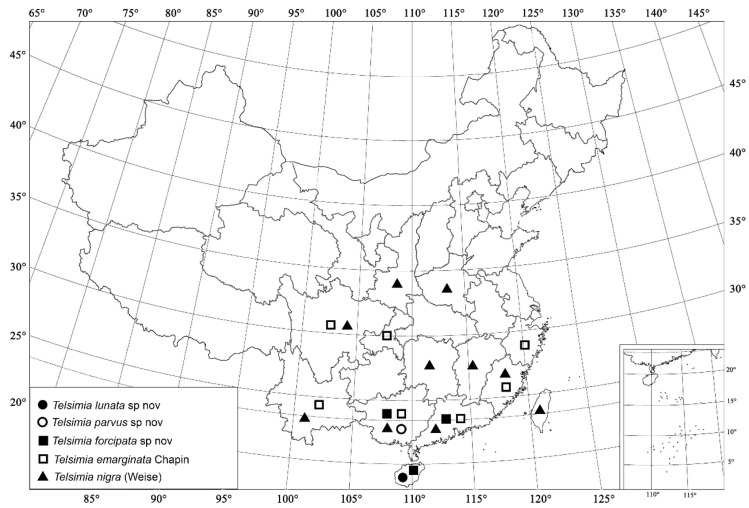
Distribution map. (●) *Telsimia lunata* sp. nov.; (○) *Telsimia parvus* sp. nov.; (■) *Telsimia forcipata* sp. nov.; (□) *Telsimia emarginata* Chapin; (▲) *Telsimia nigra* (Weise).

**Figure 9 insects-13-00869-f009:**
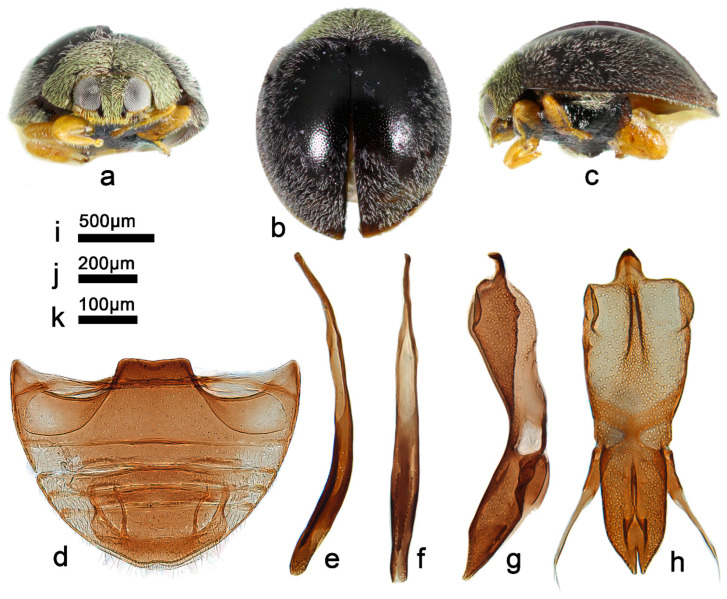
*Telsimia parvus* sp. nov. (**a**–**c**) Adult: frontal, dorsal, and lateral view; (**d**) Abdomen, male; (**e**) penis, lateral view; (**f**) penis, ventral view; (**g**) penis, lateral view; (**h**) penis guide, ventral view. **Scale bars:** Adult—(**i**), abdomen—(**j**), male genitalia—(**k**).

**Figure 10 insects-13-00869-f010:**
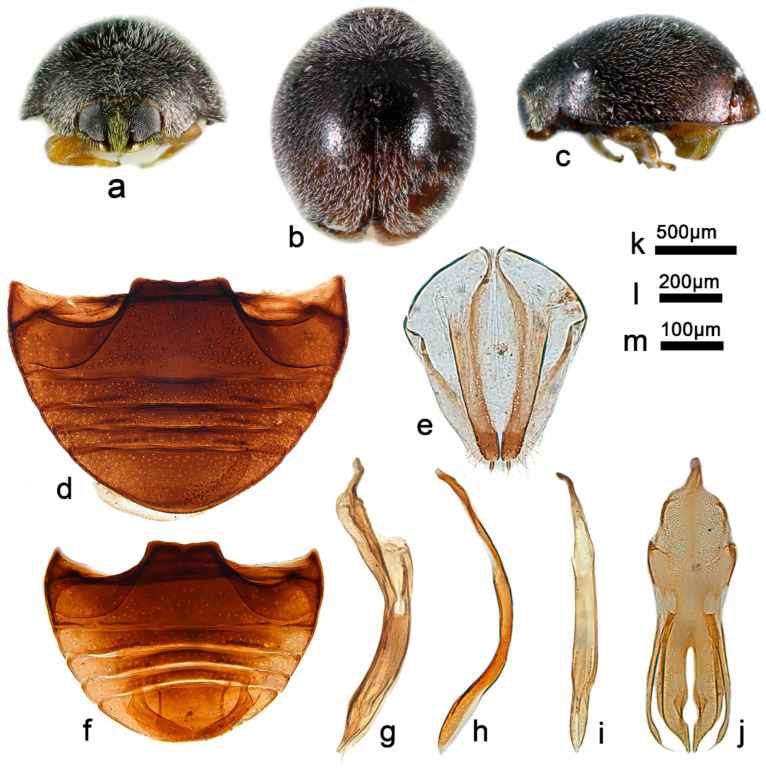
*Telsimia forcipata* sp. nov. (**a**–**c**) Adult: frontal, dorsal, and lateral view; (**d**) Abdomen, female; (**e**) ovipositor; (**f**) abdomen, male; (**g**) penis guide, lateral view; (**h**) penis, lateral view; (**i**) penis, ventral view; (**j**) penis guide, ventral view. **Scale bars:** Adult—(**k**), abdomen—(**l**), bisexual genitalia—(**m**).

**Figure 11 insects-13-00869-f011:**
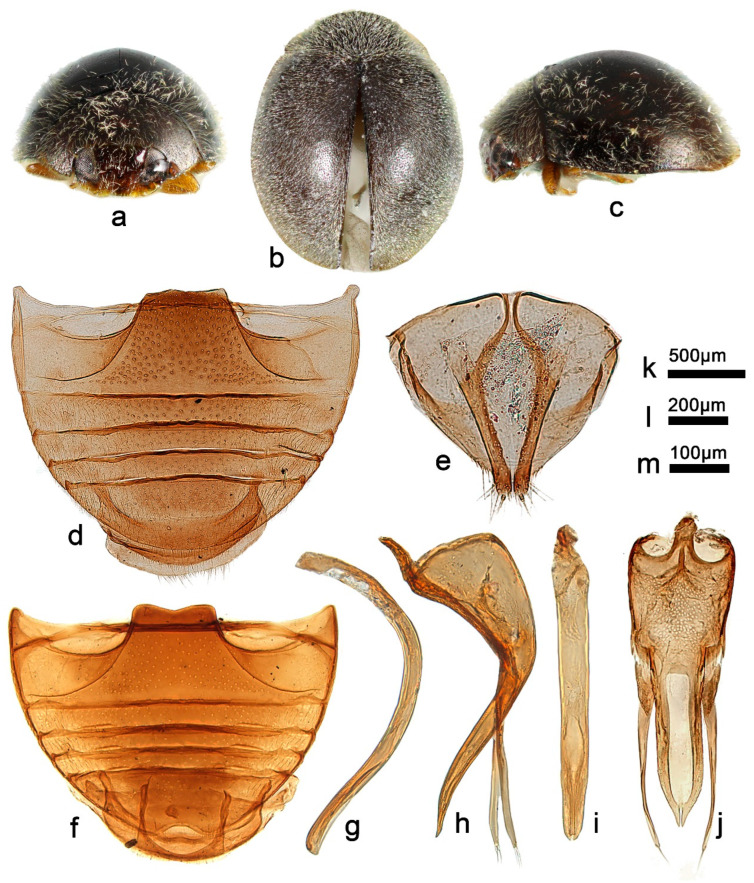
*Telsimia emarginata* Chapin. (**a**–**c**) Adult: frontal, dorsal, and lateral view; (**d**) abdomen, female; (**e**) ovipositor; (**f**) abdomen, male; (**g**) penis, lateral view; (**h**) penis guide, lateral view; (**i**) penis, ventral view; (**j**) penis guide, ventral view. **Scale bars:** Adult—(**k**), abdomen—(**l**), bisexual genitalia—(**m**).

**Figure 12 insects-13-00869-f012:**
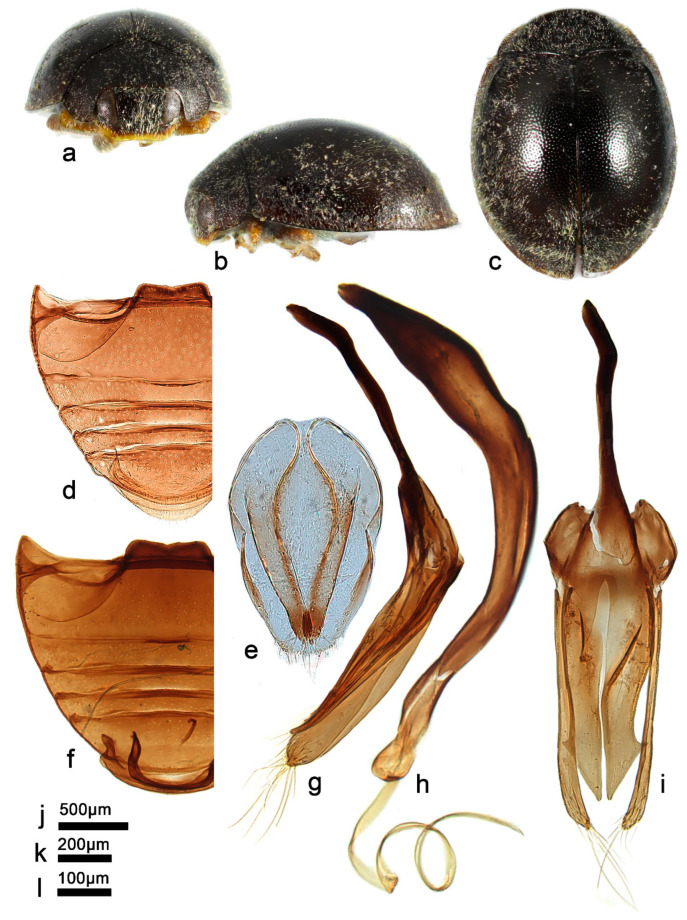
*Telsimia nigra* (Weise). (**a**–**c**) Adult: frontal, lateral, and dorsal view; (**d**) Abdomen, female; (**e**) ovipositor; (**f**) abdomen, male; (**g**) penis, lateral view; (**h**) penis guide, lateral view**;** (**i**) penis guide, ventral view. **Scale bars:** Adult—(**j**), abdomen—(**k**), bisexual genitalia—(**l**).

**Figure 13 insects-13-00869-f013:**
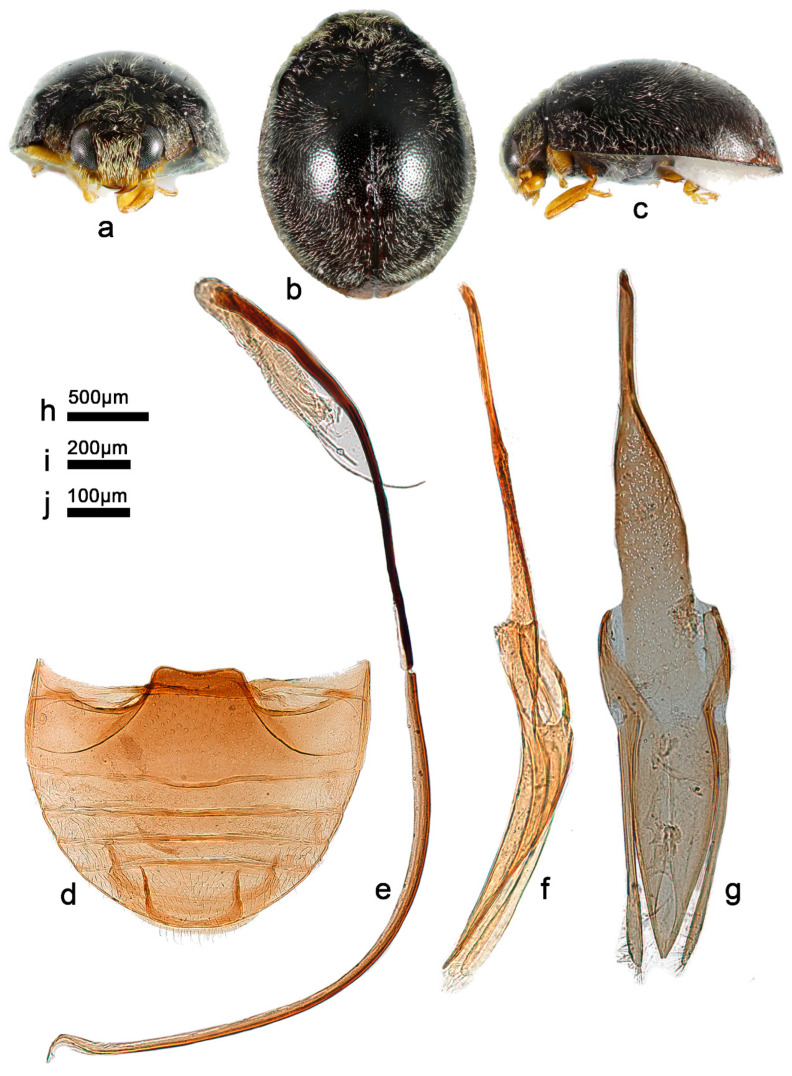
*Telsimia darjeelingensis* Kapur. (**a**–**c**) Adult: frontal, dorsal, and lateral view; (**d**) Abdomen, male; (**e**) penis, lateral view; (**f**) penis guide, lateral view; (**g**) penis guide, ventral view. **Scale bars:** Adult—(**h**), abdomen—(**i**), male genitalia—(**j**).

**Figure 14 insects-13-00869-f014:**
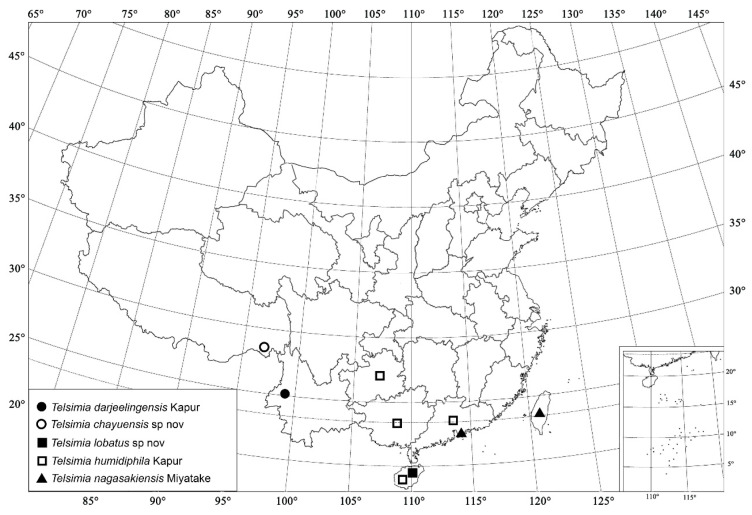
Distribution map. (●) *Telsimia darjeelingensis* Kapur; (○) *Telsimia chayuensis* sp. nov.; (■) *Telsimia lobatus* sp. nov.; (□) *Telsimia humidiphila* Kapur; (▲) *Telsimia nagasakiensis* Miyatake.

**Figure 15 insects-13-00869-f015:**
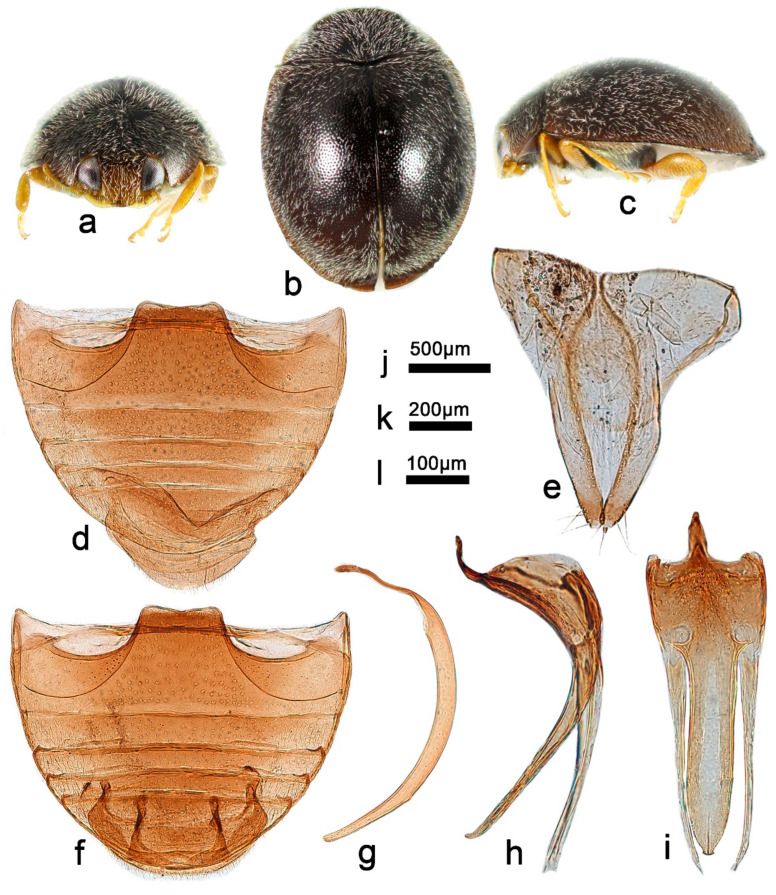
*Telsimia chayuensis* sp. nov. (**a**–**c**) Adult: frontal, dorsal, and lateral view; (**d**) Abdomen, female; (**e**) ovipositor; (**f**) abdomen, male; (**g**) penis, lateral view; (**h**) penis guide, lateral view; (**i**) penis guide, ventral view. **Scale bars:** Adult—(**j**), abdomen—(**k**), bisexual genitalia—(**l**).

**Figure 16 insects-13-00869-f016:**
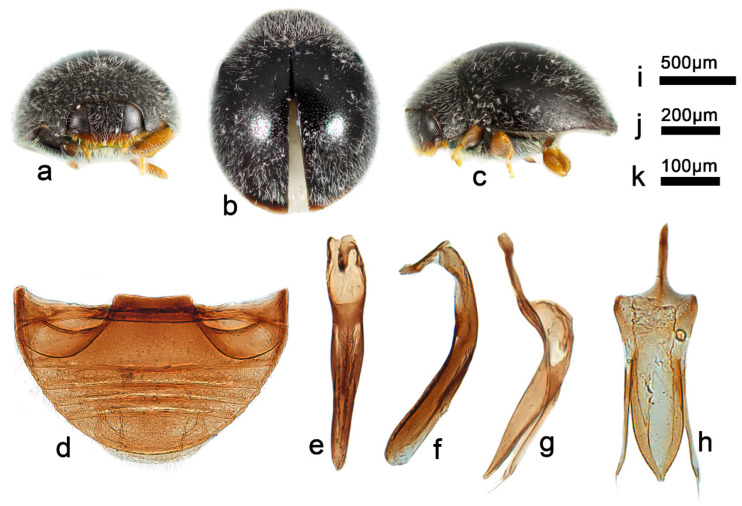
***Telsimia lobatus* sp.** nov. (**a**–**c**) Adult: frontal, dorsal, and lateral view; (**d**) Abdomen, male; (**e**) penis, ventral view; (**f**) penis, lateral view; (**g**) penis guide, lateral view**;** (**h**) penis guide, ventral view. **Scale bars:** Adult—(**i**), abdomen—(**j**), male genitalia—(**k**).

**Figure 17 insects-13-00869-f017:**
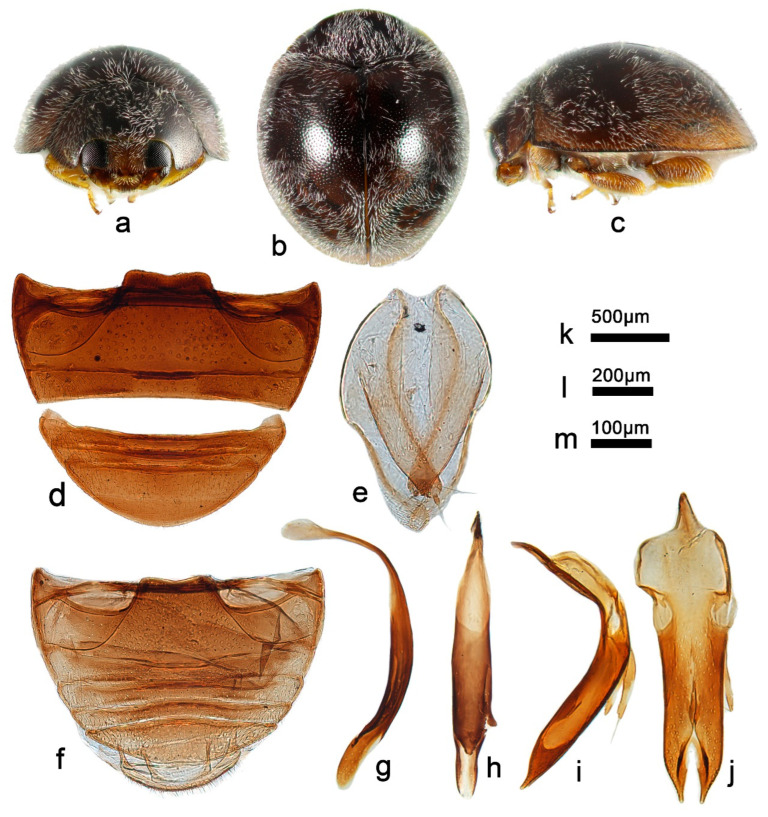
*Telsimia humidiphila* Kapur. (**a–c**) Adult: frontal, dorsal, and lateral view; (**d**) Abdomen, female; (**e**) ovipositor; (**f**) abdomen, male; (**g**) penis, lateral view; (**h**) penis, ventral view; (**i**) penis guide, lateral view**;** (**j**) penis guide, ventral view; **Scale bars:** Adult—(**k**), abdomen—(**l**), bisexual genitalia—(**m**).

**Figure 18 insects-13-00869-f018:**
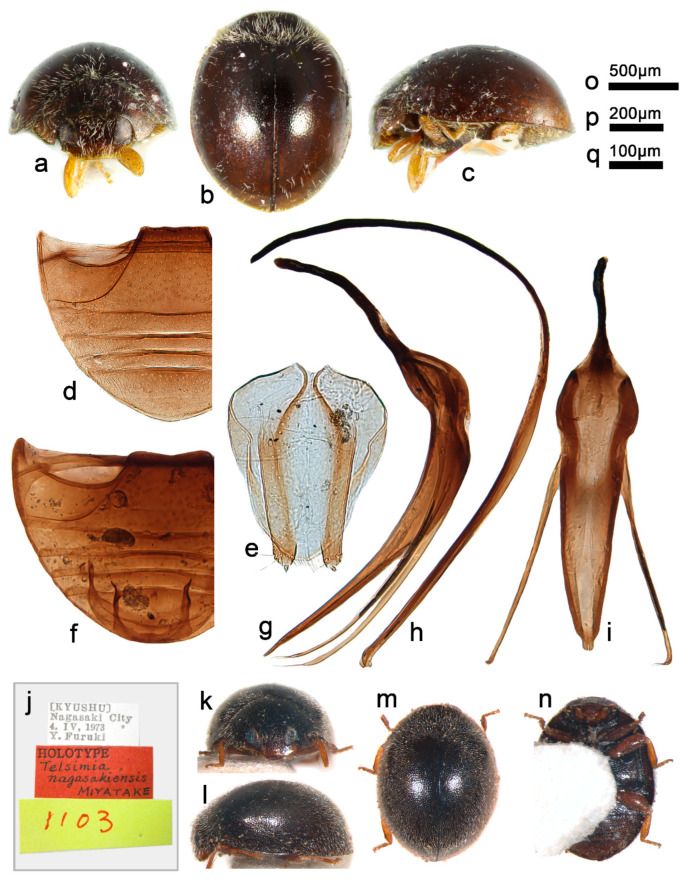
*Telsimia nagasakiensis* Miyatake. (**a–c**) Adult: frontal, dorsal, and lateral view; (**d**) Abdomen, female; (**e**) ovipositor; (**f**) abdomen, male; (**g**) penis guide, lateral view**;** (**h**) penis, lateral view; (**i**) penis guide, ventral view; (**j–n**) Holotype**. Scale bars:** a, b, c—(**o**), abdomen—(**p**), bisexual genitalia—(**q**).

**Figure 19 insects-13-00869-f019:**
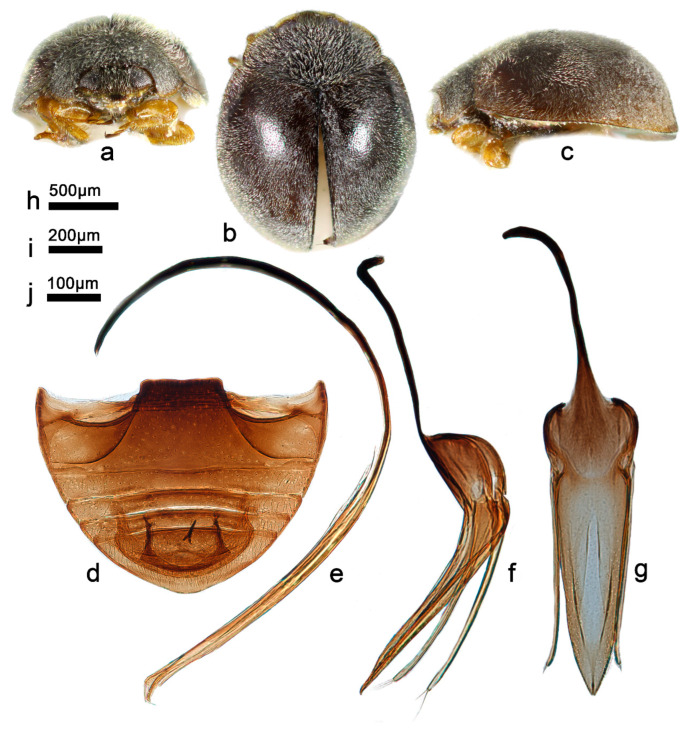
*Telsimia chujoi* Miyatake. (**a–c**) Adult: frontal, dorsal, and lateral view; (**d**) Abdomen, male; (**e**) penis, lateral view; (**f**) penis guide, lateral view**;** (**g**) penis guide, ventral view. **Scale bars:** Adult—(**h**), abdomen—(**i**), male genitalia—(**j**).

**Figure 20 insects-13-00869-f020:**
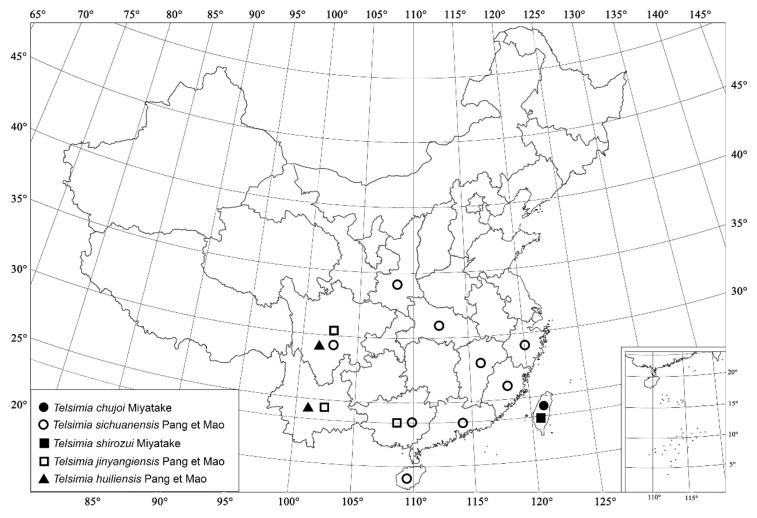
Distribution map. (●) *Telsimia chujoi* Miyatake; (○) *Telsimia sichuanensis* Pang and Mao; (■) *Telsimia shirozui* Miyatake; (□) *Telsimia jinyangiensis* Pang and Mao; (▲) *Telsimia huiliensis* Pang and Mao.

**Figure 21 insects-13-00869-f021:**
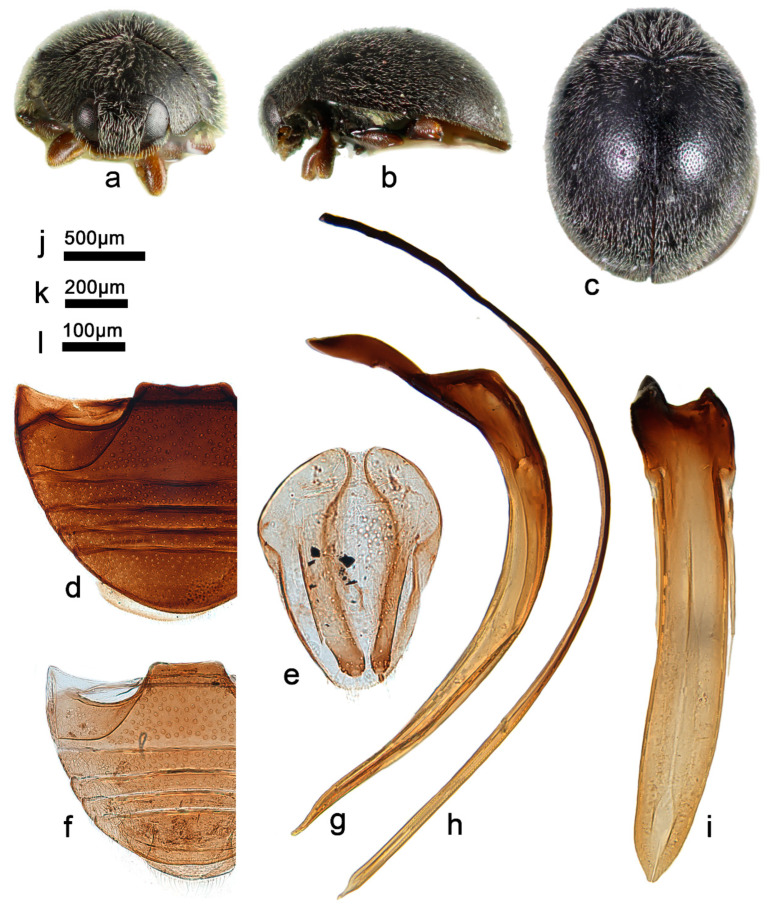
*Telsimia sichuanensis* Pang and Mao. (**a–c**) Adult: frontal, lateral, and dorsal view; (**d**) abdomen, female; (**e**) ovipositor; (**f**) abdomen, male; (**g**) penis guide, lateral view; (**h**) penis, lateral view; (**i**) penis guide, ventral view; **Scale bars:** Adult—(**j**), abdomen—(**k**), bisexual genitalia—(**l**).

**Figure 22 insects-13-00869-f022:**
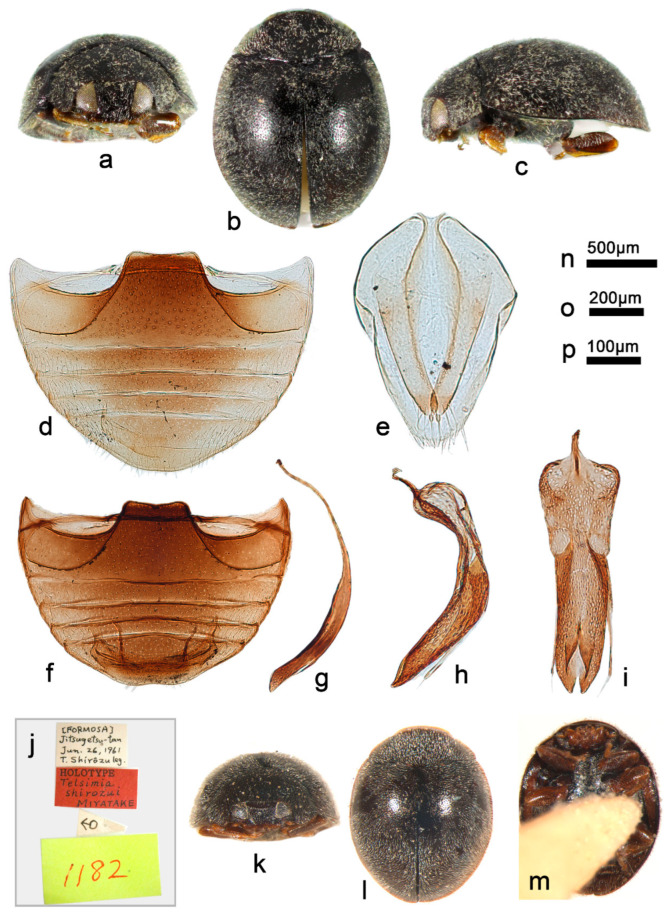
*Telsimia shirozui* Miyatake. (**a–c**) Adult: frontal, dorsal, and lateral view; (**d**) abdomen, female; (**e**) ovipositor; (**f**) abdomen, male; (**g**) penis, lateral view; (**h**) penis guide, lateral view; (**i**) penis guide, ventral view; (**j–m**) Holotype**. Scale bars:** a, b, c—(**n**), abdomen—(**o**), bisexual genitalia—(**p**).

**Figure 23 insects-13-00869-f023:**
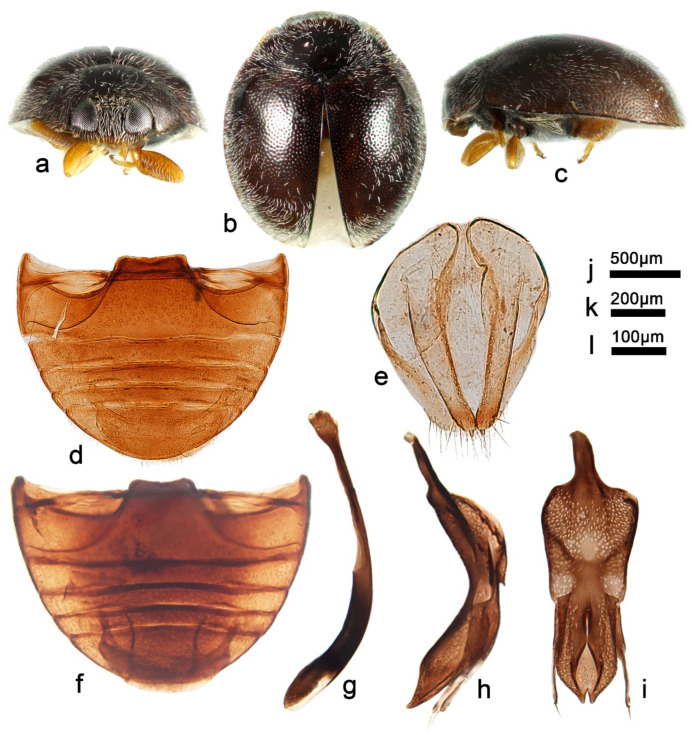
*Telsimia jinyangiensis* Pang and Mao. (**a–c**) Adult: frontal, dorsal, and lateral view; (**d**) Abdomen, female; (**e**) ovipositor; (**f**) abdomen, male; (**g**) penis, lateral view; (**h**) penis guide, lateral view; (**i**) penis guide, ventral view; **Scale bars:** Adult—(**j**), abdomen—(**k**), bisexual genitalia—(**l**).

**Figure 24 insects-13-00869-f024:**
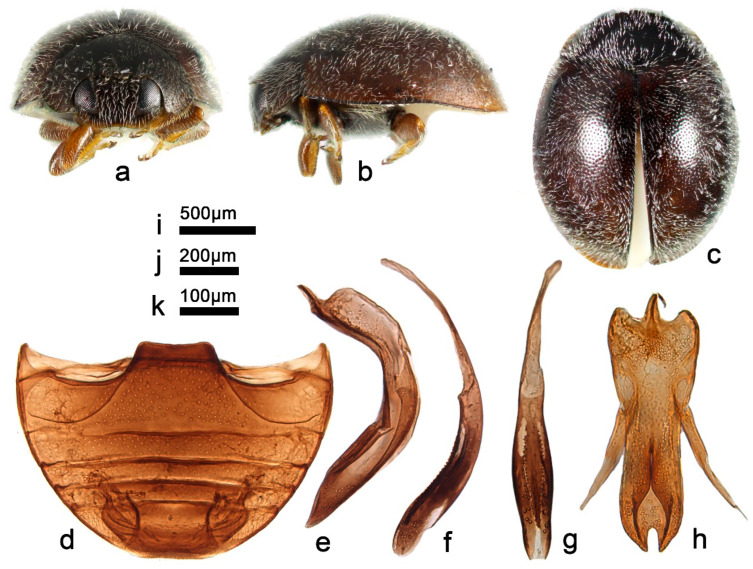
*Telsimia huiliensis* Pang and Mao. (**a–c**) Adult: frontal, lateral, and dorsal view; (**d**) Abdomen, male; (**e**) penis guide, lateral view**;** (**f**) penis, lateral view; (**g**) penis, ventral view; (**h**) penis guide, ventral view. **Scale bars:** Adult—(**i**), abdomen—(**j**), male genitalia—(**k**).

## Data Availability

All data are available in this paper.

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
