# Peer review of "A Taxonomic Review of the Genus Telsimia Casey (Coleoptera, Coccinellidae) from China, with Descriptions of Eight New Species"

_insects, 2022, doi:10.3390/insects13100869_

Round 1
Reviewer 1 Report
In this paper, 20 species of the genus Telsimia from China are reviewed, including eight new species and two new records. This taxonomic study can provide a theoretical basis for the development and utilization of Telsimia species for biological control of diaspidine scale insects. I enjoyed to read this paper. I would be happy to suggest to accept the MS after minor revision. Before formally accepted, there are still several issues should be addressed.
1. Some figures should be revised. E.g. Fig 7K and 7J are overlapped. It is better to show them all.
2. Fig 8b, 8f, there are still some shadows. Please remove the background.
3. It is better to add some arrows to highlight the important characters which were mentioned in the keys. It would be helpful for the readers.
Author Response
- Some figures should be revised. E.g. Fig 7K and 7J are overlapped. It is better to show them all.
Answer: Accepted.
- Fig 8b, 8f, there are still some shadows. Please remove the background.
Answer: Accepted.
- It is better to add some arrows to highlight the important characters which were mentioned in the keys. It would be helpful for the readers.
Answer: Accepted. According to the comments of reviewer 2, we have recompiled the key based on the external morphological characters instead of male genitalia characters.
Reviewer 2 Report
The manuscript titled “A Taxonomic Review of the Genus Telsimia Casey (Coleoptera, Coccinellidae) from China” refers to Telsimia, genus of Telsimini, which contains small beetles, insect predators with the potential as biocontrol agents. The manuscript contains elements typical for studies in the field of ​​alpha taxonomy. The Chinese twenty members of the Coccinellid genus Telsimia are reviewed, among them eight are described as new to science. Nomenclatural history, diagnoses, detailed morphological descriptions (of new for science species only), and distributions are provided for species. The graphics look very correct with color photos of adult, male and female genitalia, and distribution map, so are valuable aspect of the paper. The Telsimia fauna in China is poorly known, thus all new or reviewed information presented in this work is needed and valuable. The manuscript could be published but after considering, the following suggestions:
1. Title: line 2: in the title should be an information about new to science species described in this paper
2. Introduction, line 42-43: “Studies on Chinese Telsimia species are relatively scarce, and this knowledge gap should be filled”. Please specify that enigmatic thought. The authors could provide the state of knowledge about the presented species, what is known (about habitat or food preferences), together with literature sources. Although extensive literature is given earlier (line 41-42), it is difficult to say which items apply to the works on the Chinese Telsimia.
3. Results: Taxonomy
Generally: In the characteristics of each species, information of their habitats and biology should be supplemented as far as possible. That data (about e.g. T. nagasakiensis, T. emarginata etc.) could be easily found in the literature given by the Authors in the chapter References.
Results: Key to the Asian Species of the Genus Telsimia Casey
The key mainly based on the characteristics of the male genitalia is very controversial. It is certain that male genitalia are strong data in species diagnostics, but as the only feature (in relation to most of the presented species) it raises doubts, as to the validity and usefulness, because:
- the key completely doesn't work with regard to females,
- as it results from the analysis of the morphological characters presented by Authors in the "description" of all species new to science, some of these characters may be useful in species diagnostics and building a key, e.g.: shape of the body, length to breadth of elytra, kind of pubescence, etc.,
- line 38-40: ”This taxonomic study can provide a theoretical basis for the development and utilization of Telsimia species for biological control of diaspidine scale insects”. It's a bit hard to agree with this statement. If species identification is based solely on the male genitalia, then it is imperative to kill the individuals. This method of determination is not suitable for developmental studies where live samples are needed.
- In the paper: “Revision of the Australian Coccinellidae (Coleoptera). Part 4. Tribe Telsimini” ÅšlipiÅ„ski et al. (2005.), the authors provide a comprehensive key to the sixteen Australian species of Telsimia considering solely on morphological structures. Characters of male genitals were used only as auxiliaries and only for four species. Therefore, the Authors of the presented work are strongly encouraged to search for such morphological features among Chinese species.
Author Response
- Title: line 2: in the title should be an information about new to science species described in this paper
Answer: Accepted. The title has been changed to “A Taxonomic Review of the Genus Telsimia Casey (Coleoptera, Coccinellidae) from China, with descriptions of eight new species”
- Introduction, line 42-43: “Studies on Chinese Telsimia species are relatively scarce, and this knowledge gap should be filled”. Please specify that enigmatic thought. The authors could provide the state of knowledge about the presented species, what is known (about habitat or food preferences), together with literature sources. Although extensive literature is given earlier (line 41-42), it is difficult to say which items apply to the works on the Chinese Telsimia.
Answer: Accepted. “Studies on Chinese Telsimia species are relatively scarce, and this knowledge gap should be filled”. In this sentence, we would like to express that there are relatively few taxonomic studies on Chinese Telsimia species. Since 1979, when Pang and Mao published four new species from China, no new species have been described from China. With the accumulation of ladybug specimens in recent years, it is necessary to carry out more taxonomic studies on Chinese Telsimia species. We have revised the corresponding expressions in the manuscript.
- Results: Taxonomy
Generally: In the characteristics of each species, information of their habitats and biology should be supplemented as far as possible. That data (about e.g. T. nagasakiensis, T. emarginata etc.) could be easily found in the literature given by the Authors in the chapter References.
Answer: Accepted. In the results, we added a paragraph “Biology” for each species and six of them were provided biological information such as the prey.
Results: Key to the Asian Species of the Genus Telsimia Casey
The key mainly based on the characteristics of the male genitalia is very controversial. It is certain that male genitalia are strong data in species diagnostics, but as the only feature (in relation to most of the presented species) it raises doubts, as to the validity and usefulness, because:
- the key completely doesn't work with regard to females,
- as it results from the analysis of the morphological characters presented by Authors in the "description" of all species new to science, some of these characters may be useful in species diagnostics and building a key, e.g.: shape of the body, length to breadth of elytra, kind of pubescence, etc.,
- line 38-40: ”This taxonomic study can provide a theoretical basis for the development and utilization of Telsimia species for biological control of diaspidine scale insects”. It's a bit hard to agree with this statement. If species identification is based solely on the male genitalia, then it is imperative to kill the individuals. This method of determination is not suitable for developmental studies where live samples are needed.
- In the paper: “Revision of the Australian Coccinellidae (Coleoptera). Part 4. Tribe Telsimini” ÅšlipiÅ„ski et al. (2005.), the authors provide a comprehensive key to the sixteen Australian species of Telsimia considering solely on morphological structures. Characters of male genitals were used only as auxiliaries and only for four species. Therefore, the Authors of the presented work are strongly encouraged to search for such morphological features among Chinese species.
Answer: Accepted. We have recompiled the key based on the external morphological characters instead of male genitalia characters. At the same time, the key to Asian species is modified to the key to Chinese species, because some species are not available.
Other modifications:
The species were reordered according to the new key;
All figures have been reformatted;
The order of references;
Other small changes.
Round 2
Reviewer 2 Report
I would like to thank the Authors for their positive attitude to my comments and for improving the mnuscript.